# A G-protein pathway determines grain size in rice

Shengyuan Sun[1], Lei Wang[1], Hailiang Mao[1], Lin Shao[1], Xianghua Li[1], Jinghua Xiao[1], Yidan Ouyang[1] & Qifa Zhang[1]

Manipulating grain size is an effective strategy for increasing cereal yields. Here we identify a pathway composed of five subunits of the heterotrimeric G proteins that regulate grain length in rice. The Gβ protein is essential for plant survival and growth. Gα provides a foundation for grain size expansion. Three Gγ proteins, DEP1, GGC2 and GS3, antagonistically regulate grain size. DEP1 and GGC2, individually or in combination, increase grain length when in complex with Gβ. GS3, having no effect on grain size by itself, reduces grain length by competitively interacting with Gβ. By combining different G-protein variants, we can decrease grain length by up to 35% or increase it by up to 19%, which leads to over 40% decreasing to 28% increasing of grain weight. The wide existence of such a conserved system among angiosperms suggests a possible general predictable approach to manipulating grain/organ sizes.

---

[1] National Key Laboratory of Crop Genetic Improvement and National Centre of Plant Gene Research (Wuhan), Huazhong Agricultural University, Wuhan 430070, China. Correspondence and requests for materials should be addressed to Y.O. (email: diana1983941@mail.hzau.edu.cn) or to Q.Z. (email: qifazh@mail.hzau.edu.cn)

Grain size is a major determinant of grain weight, which is a yield component trait for cereals. Grain size is also a trait for grain quality focused by rice breeders, as long and slender grains are preferred by rice consumers in many countries. Recent advances in rice functional genomics facilitated the cloning of a series of loci controlling grain size, including genes for grain length such as GS3[1], GL3.1[2,3], An-1[4], GLW7[5] and GS2[6,7]; genes for grain width such as GW2[8], GW5[9,10], GS5[11], GW8[12] and GW7[13,14]; and genes for grain weight such as GIF1[15], GE[16], TGW6[17], GW6a[18], BG1[19] and XIAO[20].

G proteins are guanine nucleotide-binding trimeric proteins consisting of Gα, Gβ and Gγ subunits, and are involved in transmitting signals from a variety of stimuli outside a cell to its interior, thereby regulating various biological processes both in animals and plants. In animals, there are 16 Gα, 5 Gβ, 14 Gγ proteins and a large number of agonist-bound G-protein coupled receptors (GPCRs)[21]. In mammalian cells, G protein signaling is initiated by GPCRs, such that the Gα subunit is activated through the exchange of a GDP for GTP, which causes its dissociation with the Gβγ dimer. The disassociated Gα and Gβγ activate their own downstream effectors respectively thus performing various biological functions[22-24]. Most plant genomes contain much fewer G proteins[23,25], and plant G proteins are self-activating without involving GPCR[24]. However, biological functions of plant G proteins have been much less studied.

Rice encodes one each of Gα and Gβ, and five Gγ proteins[25,26]. Both Gα and Gβ proteins are positive regulators of cell proliferation and grain size growth[27-30]. Gγ proteins are divided into three distinct groups according to their C-terminal structures[31]. Two typical Gγ subunits RGG1 (Group I) and RGG2 (Group II) are involved in regulation of abiotic stresses[32-34]. Natural variants of an atypical Gγ subunit DEP1 (Group III) showed functions in panicle architecture and nitrogen-use efficiency[35-37]. GS3, encoding a Gγ subunit (Group III) of the heterotrimeric G proteins, is a major QTL for grain size[1,38]. The wild-type GS3 protein produces medium grain, a loss-of-function allele results in long grain, while a truncated form lacking the C-terminus produces very short grain[39]. Numerous studies have established that GS3 is the most important regulator of grain length in both natural and breeding populations of cultivated rice[40-42] and its natural variants have contributed greatly to the improved productivity and grain quality of rice in a global scale. The Gα, Gβ and Gγ proteins were also found to influence organ size and shape in Arabidopsis[43-46], suggesting a general function of G protein signaling in organ size regulation in plants.

In this study, we identify a pathway made up of five subunits of the heterotrimeric G proteins that regulates grain size in rice. We show that manipulating the three Gγ proteins, DEP1, GGC2 and GS3, can achieve designed grain size, demonstrating a predictable approach to improve grain yield and quality.

## Results

### GS3 and DEP1 conferred divergent functions on grain size.
The GS3 protein shares significant similarity with DEP1 in the N-terminal region, which was named OSR (organ size regulation) domain[39] and later recognized as Gγ-like domain[26]. The OSR domain of GS3 showed 68.7% identity to DEP1 at the DNA level and 50% identity by protein sequence. A number of natural variants of DEP1 and GS3 were found showing differences mostly in the length and composition of the C-terminal cysteine-rich domain[36,39,47,48] (referred to as "tail" hereafter) (Fig. 1). DEP1 has a long tail of 305 amino acids in length; a 625-bp deletion in DEP1 causes a premature stop codon[35,36], producing a truncated protein dep1 with a tail of 75 amino acids. The wild-type GS3 allele from the rice variety Zhenshan 97 (GS3-1) contains a

relative short tail (115 amino acids). The GS3-4 allele from a rice variety Chuan 7 encodes a truncated GS3 protein lacking the entire tail and containing the OSR domain only[39].

To compare the effects of these alleles on grain size, we transformed Zhonghua 11 (ZH11), a rice variety harboring functional alleles of both DEP1 and GS3-1 with medium grain size, with constructs overexpressing DEP1 (DEP1OE), dep1 (dep1OE), GS3-1 (GS3-1OE) and GS3-4 (GS3-4OE) respectively, and RNA-interference of DEP1 (DEP1Ri) and GS3-1 (GS3-1Ri). Transcript levels of all genes were checked in the young panicles of the transformants (Supplementary Fig. 1a, b).

We field-tested 2 or 3 T₁ families randomly selected from T₀ plants. Compared with the negative segregants, elevated DEP1 accumulation increased the grain length by 6.85–9.58% with a normal plant stature. However, plants overexpressing dep1 showed very similar phenotype to DEP1Ri plants, both of which reduced grain length by ~4.5%, together with dwarf stature and erect panicles (Fig. 1a and Supplementary Table 1). The expression level of endogenous DEP1 in dep1OE was not reduced (Supplementary Fig. 1c). Thus dep1 showed a dominant-negative effect over DEP1 in regulating grain size rather than co-suppression of the two genes. In contrast to DEP1OE, GS3-1OE showed an average 9.07% reduction in grain length together with reduced plant stature, very similar to the phenotype of dep1OE (Fig. 1b and Supplementary Table 1). GS3-4OE produced even smaller plants and grain size with an average 19.10% reduction of grain length relative to the negative segregant, while GS3-1Ri resulted in an average 5.78% increase of grain length. These results were similar to those of Mao et al[39].

Because the major difference of DEP1, dep1 and GS3 lies in the C-terminal domain, the phenotype differences of the transgenic plants suggested that the tail length and composition of these genes are important for their functions. DEP1 with a long tail had a positive effect on grain length, while dep1 and GS3-1 with short tails played negative roles in grain length. Complete loss of the tail in GS3-4 further enhanced the negative effect of GS3 in grain length regulation.

### Interaction between DEP1 and GS3 in grain size regulation.
We crossed GS3-1Ri with dep1OE plants, all GS3-1Ri/dep1OE F₁ plants exhibited reduced grain length, similar to dep1OE transgenic plants. In the F₂ generation, all the GS3-1Ri/dep1OE plants also showed the dep1OE phenotype with respect to grain size, plant height and panicle length (Supplementary Fig. 2a).

Next, we introduced GS3-1 and dep1 driven by their own native promoters into a T-DNA mutant of GS3 with increased grain size (Supplementary Fig. 2b-c). Both transformants produced significant shorter grains, again supporting the conclusion that GS3-1 and dep1 had similar role in grain size regulation.

Using CRISPR/Cas9 editing, we obtained two independent T₁ families for each of GS3-1 (GS3-1ko) and DEP1 (DEP1ko) single and double (GS3-1koDEP1ko) mutants in the background of ZH11 (Fig. 2a and Supplementary Fig. 3a–d). Consistent with the observation in RNAi plants, homozygous GS3-1ko mutant increased grain size, whereas homozygous DEP1ko mutant reduced grain size. The grain length of GS3-1koDEP1ko double mutant was intermediate between that of the single mutants of GS3-1ko and DEP1ko. To confirm the result, we crossed a GS3-1koDEP1ko double mutant with ZH11. In the F₂ segregants, GS3-1ko homozygous plants also increased grain length, the DEP1ko homozygote reduced the grain length, while the grain length of GS3-1koDEP1ko double mutant was intermediate between that of the GS3-1ko and DEP1ko single mutants.

We further crossed GS3-1OE and GS3-4OE with DEP1ko plants. It was shown that knocking out DEP1 in the GS3-1OE

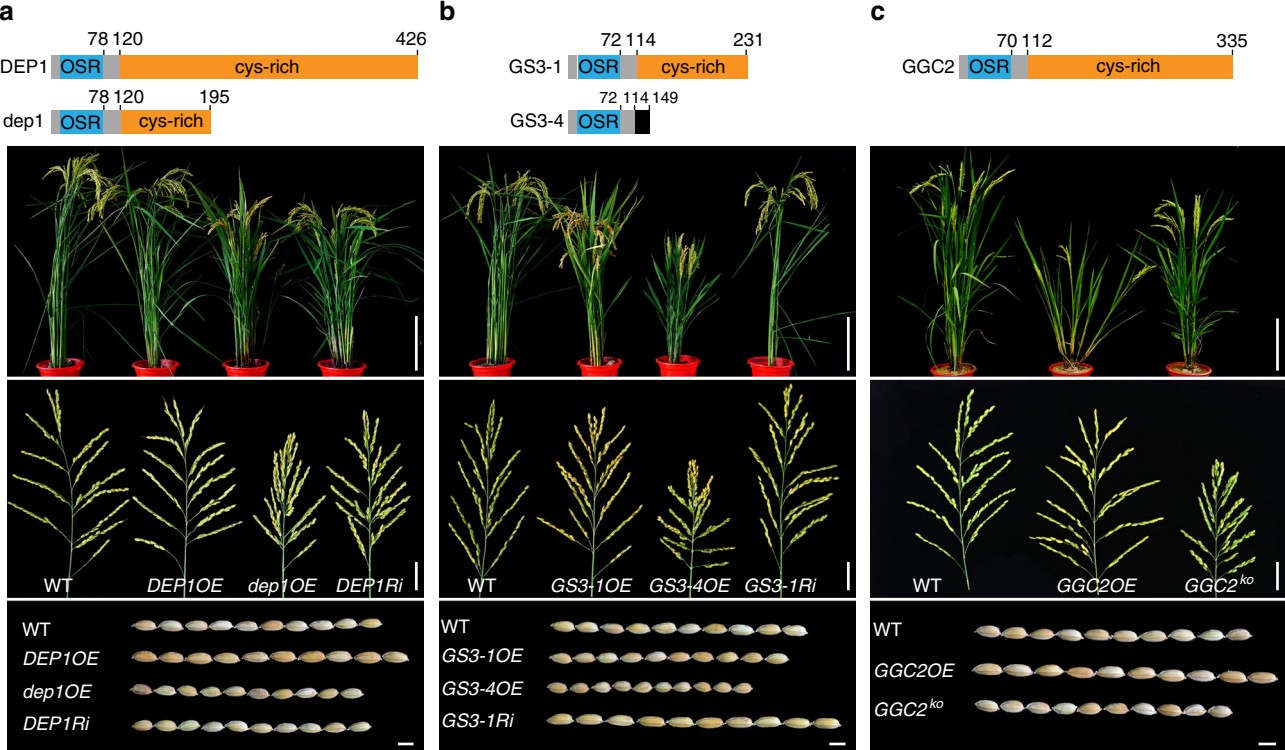

**Fig. 1** Genetic effects of three Gγ proteins on grain size and plant morphology. The structures of the proteins are shown on top of each panel, with the numbers of residues of each protein indicated. Plant, bar = 20 cm; Panicle, bar = 5 cm; grains, bar = 5 mm. **a** Top: the protein structures of DEP1 and dep1. Bottom: whole plants, panicles, and grains of WT (wild type ZH11), *DEP1OE*, *dep1OE*, and *DEP1Ri* plants. **b** Top: the protein structures of GS3-1 and GS3-4. Bottom: whole plants, panicles, and grains of WT, *GS3-1OE*, *GS3-4OE*, and *GS3-1Ri* plants. **c** Top: the protein structure of GGC2. Bottom: whole plants, panicles, and grains of WT, *GGC2OE*, and *GGC2^{ko}* plants

or *GS3-4OE* background did not further reduce the gain size of *GS3-1OE* or *GS3-4OE* (Supplementary Fig. 2d). We also crossed *DEP1OE* with *GS3-1Ri* and *GS3-1OE* transgenic plants. Further increased grain length was observed when overexpressing *DEP1* in *GS3-1Ri* background, whereas the *DEP1OE/GS3-1OE* hybrid showed the *GS3-1OE* phenotype of short grain (Supplementary Fig. 2e).

**An atypical Gγ subunit GGC2 functions additively with DEP1.**
Both DEP1 (chromosome 9) and GS3 (chromosome 3) are atypical Gγ subunits. By searching the reference genome sequences of rice[49,50], we found another atypical Gγ protein (GGC2) on chromosome 8, showing 66% and 48% identities to DEP1 and GS3, respectively. Transcripts of the three genes accumulated in the panicles, whereas *DEP1* and *GGC2* were also highly expressed in shoot apex relative to *GS3* (Supplementary Fig. 1e). Overexpression of *GGC2* (*GGC2OE*) in ZH11 increased grain length significantly, whereas knock-out mutant of *GGC2^{ko}* in ZH11 reduced the grain length (Fig. 1c and Supplementary Fig. 1d, 3a, e), suggesting that like *DEP1*, *GGC2* also acted as a positive regulator of grain length in rice.

To validate the relationship of GGC2 with DEP1 and GS3, we generated *DEP1^{ko}GGC2^{ko}* double mutant and *DEP1^{ko}GGC2^{ko}GS3-1^{ko}* triple mutant (Supplementary Fig. 3f, g). Two independent T$_1$ families for each mutant were used for further investigation. Compared to the single mutant of *DEP1^{ko}* or *GGC2^{ko}*, knocking out both of them resulted in much smaller grains than either of the single knock-out mutants, suggesting that DEP1 and GGC2 worked additively in positive regulation of grain length. Moreover, the *DEP1^{ko}GGC2^{ko}GS3-1^{ko}* triple mutant produced similar phenotype to that of *DEP1^{ko}GGC2^{ko}*, thus *GS3-*

*1^{ko}* mutant could not increase grain length when both *DEP1* and *GGC2* were knocked-out (Fig. 2a).

Putting together the data from the above genetic analyses, an interesting outcome emerged concerning the genetic effects and functions of these genes. Taking the double mutants *DEP1^{ko}GGC2^{ko}* that produced the smallest grain as the base line, adding either *DEP1* (as shown by *GGC2^{ko}*) or *GGC2* (*DEP1^{ko}*) could greatly increase grain size, and the effect on grain size increase of *GGC2* was larger than that of *DEP1*. Having both *DEP1* and *GGC2* could further increase grain size (as shown by the wide-type ZH11). Although *GS3-1^{ko}* increased grain size in the presence either or both of *DEP1* and *GGC2*, it could not increase grain size in the double mutant (*DEP1^{ko}GGC2^{ko}*). Therefore it became clear that *DEP1* and *GGC2* functioned positively in regulating grain size in an additive manner, while the role of *GS3* in grain size regulation was to repress the effects of *DEP1* and *GGC2* on increasing grain size.

**Functional dependence of the Gγ subunits on RGB1 and RGA1.** The Gβ-mediated process requires a γ subunit to form a Gβγ dimer[45,51,52]. To investigate the genetic relationship between the three Gγ proteins with Gβ subunit RGB1 in grain size regulation, we tried to knock out *RGB1* in ZH11 by CRISPR/Cas9, but failed to obtain a homozygous mutant in the T$_1$ generation, likely because of lethality of the null mutant (Supplementary Fig. 4a). Therefore, we knocked down *RGB1* by RNAi (*RGB1Ri*), and the transgenic plants with suppressed expression of *RGB1* showed reduced grain size and plant height, together with brown lamina joint and internodes (Supplementary Fig. 4b), which was similar to the previously observed phenotype of suppressing *RGB1*[30]. We crossed *DEP1OE* and *GS3-1Ri* with *RGB1Ri* transgenic plants, respectively, and overexpressed *GGC2* in *RGB1Ri*.

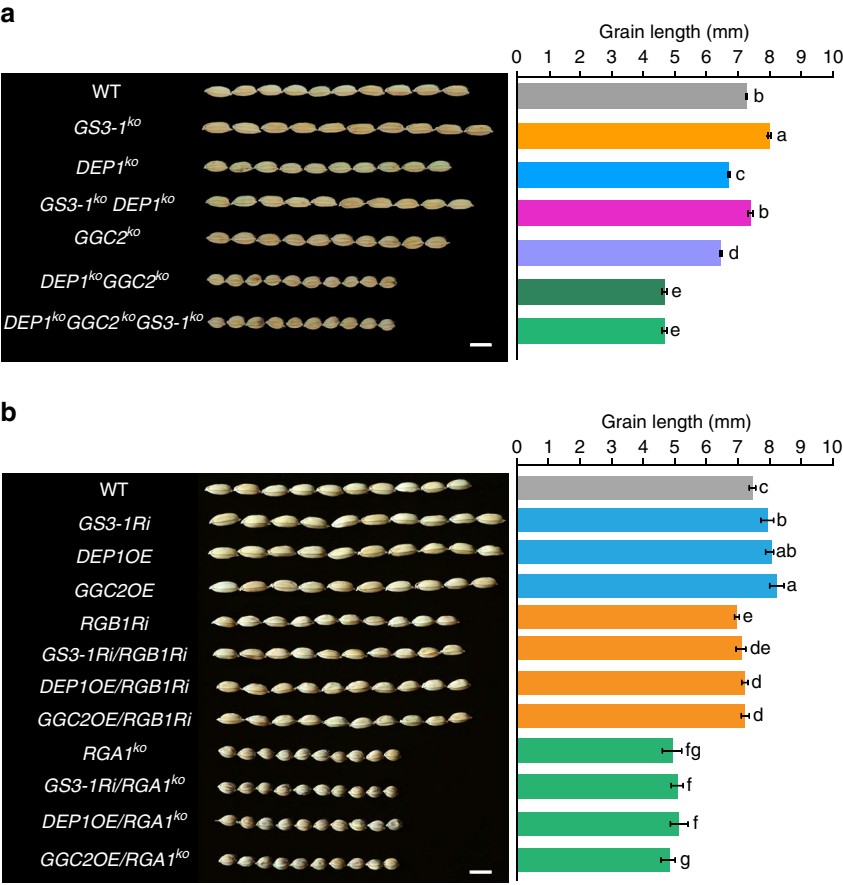

**Fig. 2** Genetic interactions of three Gγ proteins and Gβ or Gα proteins. **a** Genetic interactions of *DEP1*, *GGC2*, and *GS3*. Grains and grain length of WT (wild type ZH11) ($n = 10$), single mutants of $GS3-1^{ko}$ ($n = 13$), $DEP1^{ko}$ ($n = 13$), and $GGC2^{ko}$ ($n = 13$), double mutants of $GS3-1^{ko}DEP1^{ko}$ ($n = 10$) and $DEP1^{ko}GGC2^{ko}$ ($n = 5$), and triple mutant of $DEP1^{ko}GGC2^{ko}GS3-1^{ko}$ ($n = 5$). The data shown are mean ± SEM. Different letters indicate significant differences ranked by the Fisher's Least Significant Difference (LSD) test ($P < 0.05$). Bar = 5 mm. **b** Genetic interactions of the Gγ proteins with RGA1 and RGB1. Grains and grain length of WT, *GS3-1Ri*, *DEP1OE*, *GGC2OE*, and their hybrids with *RGB1Ri* and $RGA1^{ko}$, respectively. Values are given as mean ± SEM. Different letters indicate significant differences ranked by the LSD test ($P < 0.05$). Bar = 5 mm

The *RGB1Ri/DEP1OE* and *RGB1Ri/GGC2OE* plants showed reduced grain length compared to the wild-type. Similar result was also obtained in the *RGB1Ri/GS3-1Ri* hybrid (Fig. 2b). Thus the effects of grain length increase by *DEP1OE*, *GGC2OE* and *GS3-1Ri* were dependent on *RGB1*.

We also knocked out *RGA1* in ZH11. A homozygous $RGA1^{ko}-1$ mutant with a 1269-bp deletion produced very small grains and plant size relative to the wild-type (Supplementary Fig. 4c), which was similar to the previously observed phenotype of *rga1* (*d1*) mutant[28,29]. Three double mutants involving *RGA1* were generated including: $GS3-1Ri/RGA1^{ko}$ produced by editing *RGA1* (created an 804-bp deletion) in *GS3-1Ri*, $DEP1OE/RGA1^{ko}$ generated by editing *RGA1* (1269-bp deletion) in *DEP1OE*, and $GGC2OE/RGA1^{ko}$ that overexpressed *GGC2* in $RGA1^{ko}$ background. All three double mutants produced very small grains, similar to $RGA1^{ko}$ single mutant (Fig. 2b). Therefore, the effects of *DEP1OE*, *GGC2OE* and *GS3-1Ri* on grain size increase were also dependent on *RGA1*.

Thus the results of the genetic analyses showed that while RGB1 appears to play essential role for overall growth, RGA1 provided the baseline for the grain size regulation pathway composed of the three Gγ proteins, which may be summarized in Fig. 3a. In the presence of RGA1, DEP1 and GCC2, singly or together, function to increase grain size, while GS3 suppresses the effects of DEP1 and GCC2 to reduce grain size. Thus, relative to the wild type, knocking out *RGA1* produces extremely short

grains showing ~35% reduction in grain length, similar phenotype can be obtained when both *DEP1* and *GCC2* are knocked out (Fig. 3b). Knocking out either *DEP1* or *GCC2* results in moderate short grains (7.47 to 11.18% reduction), and overexpressing *GS3* produces similar phenotype. Conversely, overexpressing either *DEP1* or *GCC2*, or knocking out *GS3* results in moderate increase (6.85 to 13.03%) in grain length and grain weight. Further increase in grain length (~20% increase) is obtained when *DEP1* is overexpressed in the knock-down mutant of *GS3*, which also leads to large increase (28.45%) in grain weight. Based on the above results, we predict that overexpressing both *DEP1* and *GCC2* in the knock-out mutant of *GS3* has the potential to further increase grain size. Thus grain size could be manipulated with combinations of the alleles of Gγ proteins, providing a highly flexible and predictable approach to modifying grain size and increasing grain yield.

**The role of tail and competing interactions of Gγs with RGB1**. To investigate the relative abundance of the proteins, we generated transgenic plants expressing Flag tag-fused GS3-4 (no tail), GS3-1 (short tail), dep1 (short tail) and DEP1 (long tail) respectively in ZH11 driven by the ubiquitin promoter. These transgenic plants phenocopied the corresponding ones with elevated expression of these genes without the Flag tag (Fig. 4a, b). Western blot analysis showed that GS3-4:Flag protein

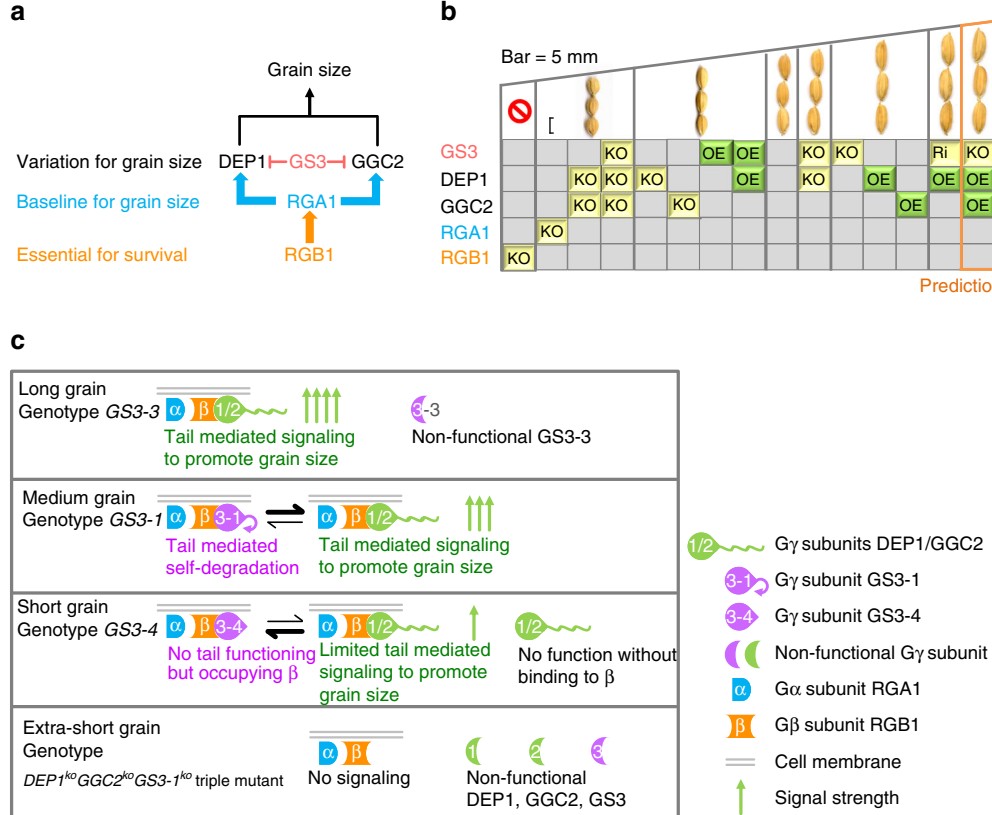

**Fig. 3** Schematic representation of the functions of the G proteins in grain size regulation. **a** A genetic model depicting the pathway of the G proteins in grain size regulation. The black arrow indicates positive effect, and the red bars represent negative effect. **b** Manipulating grain size by combinations of G proteins. KO indicates knocking out of the gene, Ri for RNAi plants, and OE for overexpression of the gene. The empty space indicates wild type. **c** A model explaining how these Gγ proteins work

accumulated in much higher amount than did GS3-1:Flag (Fig. 4c). After treating with proteasome inhibitor MG132, GS3-1:Flag protein accumulation increased to a comparable level to GS3-4:Flag (Fig. 4d), indicating different rates of degradation of the GS3-1 and GS3-4 proteins in vivo. These results suggested that the C-terminal tail of GS3 is necessary for degradation of the protein and such tail-mediated proteolysis is critical for their function in grain size regulation. However, for unknown reason, the protein could not be detected in DEP1:Flag and dep1:Flag (Fig. 4c).

Using yeast two-hybrid (Y2H), BiFC, and luciferase activity assays, we found that GS3-1, GGC2, and DEP1 interacted with RGB1 through the OSR domain (amino acid 1-94 in GS3-1) (Fig. 5a–c). In assays using the protoplast of *GS3-1OE*, *GS3-4OE* and ZH11, the interaction of RGB1 with DEP1 or GGC2 was suppressed in the *GS3-1OE* background compared with that in ZH11, suggesting that there existed a competition between GS3 and DEP1 or GGC2 in interacting with RGB1 (Fig. 5d). In addition, the interaction of RGB1 with DEP1 or GGC2 was even weaker in *GS3-4OE* background, presumably due to the over-accumulation of GS3-4 protein in transgenic plants. Such competition of GS3, DEP1 and GGC2 in interaction with RGB1 was also supported by an in vitro yeast three-hybrid assay, showing that the interaction between DEP1/GGC2 and RGB1 was disrupted by the expression of GS3, and conversely the RGB1–GS3 interaction could also be influenced by DEP1 (Fig. 5e). These results suggested that RGB1-DEP1 and RGB1-GGC2 dimers positively regulate grain size, while occupation of RGB1 by the GS3 protein disrupts such function, and over-abundant GS3 results in short grain. Such competitive interactions provide

an explanation for the antagonistic activities of these proteins in grain size regulation, allowing for the possibility that other subunits of G-proteins may also have roles in this pathway.

**Application of the Gγ system in organ size manipulation**. We searched protein sequences with significant similarity to DEP1, GS3 or GGC2 from a range of angiosperm plants, using reciprocal BLASTP (query cover > 30%, $E < e^{-4}$, identity > 30%) in the NCBI database. A total of 96 proteins were found from 55 species, including both monocot and dicot plants (Supplementary Data 1). A maximum-likelihood phylogenetic tree was constructed to depict the relatedness of these proteins, which revealed two distinct clusters (Supplementary Fig. 5). The first cluster comprised 63 proteins from dicots, and the *Arabidopsis* homolog AGG3 was in this cluster showing the highest similarity to DEP1 ($E = 2e^{-24}$). DEP1, GGC2 and GS3 belonged to the second cluster with 32 proteins from the monocots. Two maize Gγ proteins, DAA61661 (designated as ZmDEP1a, $E = 4e^{-48}$) and DAA40167 (designated as ZmDEP1b, $E = e^{-72}$), were identified showing higher similarity to DEP1 than to GGC2 and GS3. A maize Gγ protein of ACZ02400 (designated as ZmGS3, $E = 2e^{-40}$) was identified as the orthologous copy of GS3. Reciprocal blast also showed that a maize Gγ protein DAA45508 (designated as ZmGGC2, $E = 5e^{-50}$) has the highest similarity (73%) to rice GGC2. This suggested that the DEP1-GGC2-GS3 system is conserved in monocots. In addition, homologs of DEP1 also exist widely in dicots.

To investigate the effects of the various homologs on grain size regulation, we overexpressed *DEP1*, *GS3-1*, the *Arabidopsis* ortholog *AGG3*, the maize homolog *ZmGS3* and the soybean

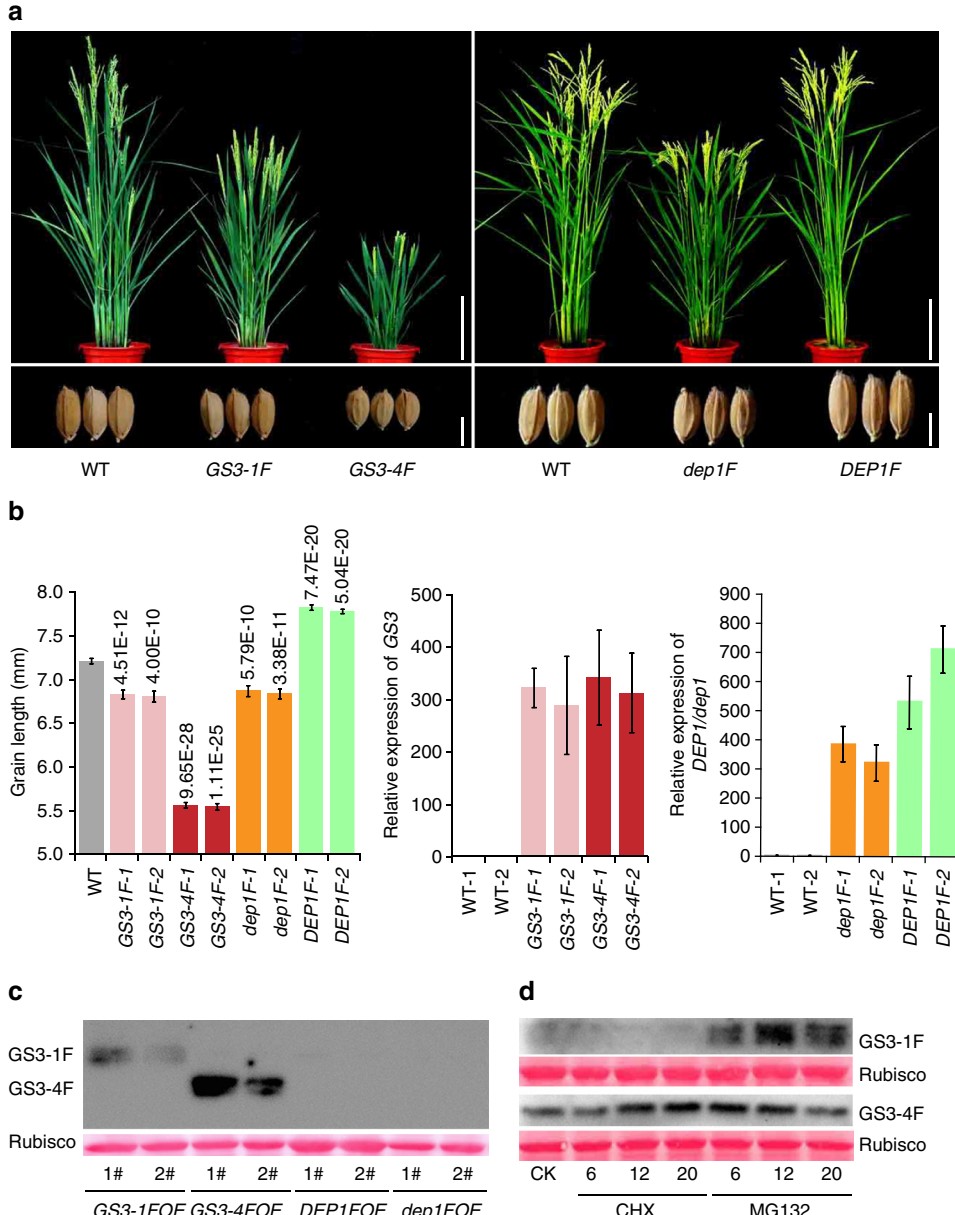

**Fig. 4** Immunoblot analysis of the proteins in transgenic plants with Flag tag. **a** Whole plants and grains in WT (wild type ZH11), and transgenic plants of *Ubi::GS3-1:Flag* (*GS3-1F*), *Ubi::GS3-4:Flag* (*GS3-4F*), *Ubi::dep1:Flag* (*dep1F*), and *Ubi::DEP1:Flag* (*DEP1F*). Plant, bar = 20 cm; grains, bar = 5 mm. **b** Grain length and relative expression levels of the genes in transgenic plants. Values for grain length are given as mean ± SEM (*n* = 10). *P* values are given on the top of the bars, which are based on two-tailed *t*-tests. Values for expression analysis are given as mean ± SEM (*n* = 3). **c** Immunoblot analysis of the transgenic plants. Total proteins are extracted from the shoots of 15-day-old seeding, and 20 μg amount was used for western blotting. **d** Immunoblot analysis of Flag fused GS3-1 and GS3-4 proteins after the treatments with 30 μM CHX and 50 μM MG132. CK, the *GS3-1F* and *GS3-4F* transgenic plants without treatment. CHX, Actidione. The two bands of GS3-1 protein are likely due to protein modification by ubiquitination. The numbers (6, 12, 20) indicate hours after the treatments

homolog *GmDEP1* in Daohuaxiang2 (DHX), a variety lacking functional *GS3*. AGG3, ZmGS3 and GmDEP1 contained short tail sequence, with 50%, 49% and 48% of protein identities to DEP1, and 45%, 74%, and 33% identities to GS3-1. The transgenic plants overexpressing *DEP1* produced expected longer grains. And overexpressing AGG3, ZmGS3 and GmDEP1 in DHX resulted in reduced plant size and grain length, similar to the overexpression of *GS3-1*, also as expected on the basis of tail length. Furthermore, overexpressing the predicted OSR domains of *AGG3* and *ZmGS3* in DHX both resulted in reduced grain length and plant size (Fig. 6a and Supplementary Table 2). Thus differences in sequences of the N-terminal OSR domain or the entire proteins

did not make difference in grain length regulation, but the tail length is critical. The long-tailed protein produced long grain, while the short-tailed and tailless proteins or the OSR domains alone produced short grain.

But the situation is different in *Arabidopsis* transformation. It was previously shown that overexpression of *AGG3* in *Arabidopsis* increased organ size[43]. Introduction of *DEP1*, *GGC2*, and *GS3-1* driven by the *35S* promoter into *agg3-2*, a knock-out mutant of *AGG3* with decreased organ size[43], recovered the phenotype showing increased flower organ, silique length, and seed size compared with the *agg3-2* mutant (Fig. 6b, c). Therefore, *Arabidopsis* did not distinguish the tail length difference of Gγ

proteins, suggesting that *Arabidopsis* may not have evolved such a mechanism to be impacted by the tail-length difference in organ size regulation.

## Discussion

Based on the results of this study, we proposed a model to explain how these Gγ proteins work in grain size regulation (Fig. 3c). In this model, DEP1 and GGC2, when coupled with RGB1, promote grain size by tail-mediated signaling. GS3, though having no function in promoting or inhibiting grain size, reduces grain size by blocking the interaction of DEP1 and GGC2 with RGB1. The tail-mediated self-degradation of GS3 in the RGB1-GS3 complex provides a dynamic balance between blocking and the DEP1/GGC2–RGB1 interaction, thus a plant carrying *GS3-1* produces medium grain. Without the tail (GS3-4), accumulated GS3-4 would largely occupy RGB1, resulting in short grain. The net outcome of the functional interactions among these genes thus determines grain size, allowing for the possibility that other subunits of G-proteins may also be involved in such a signaling pathway.

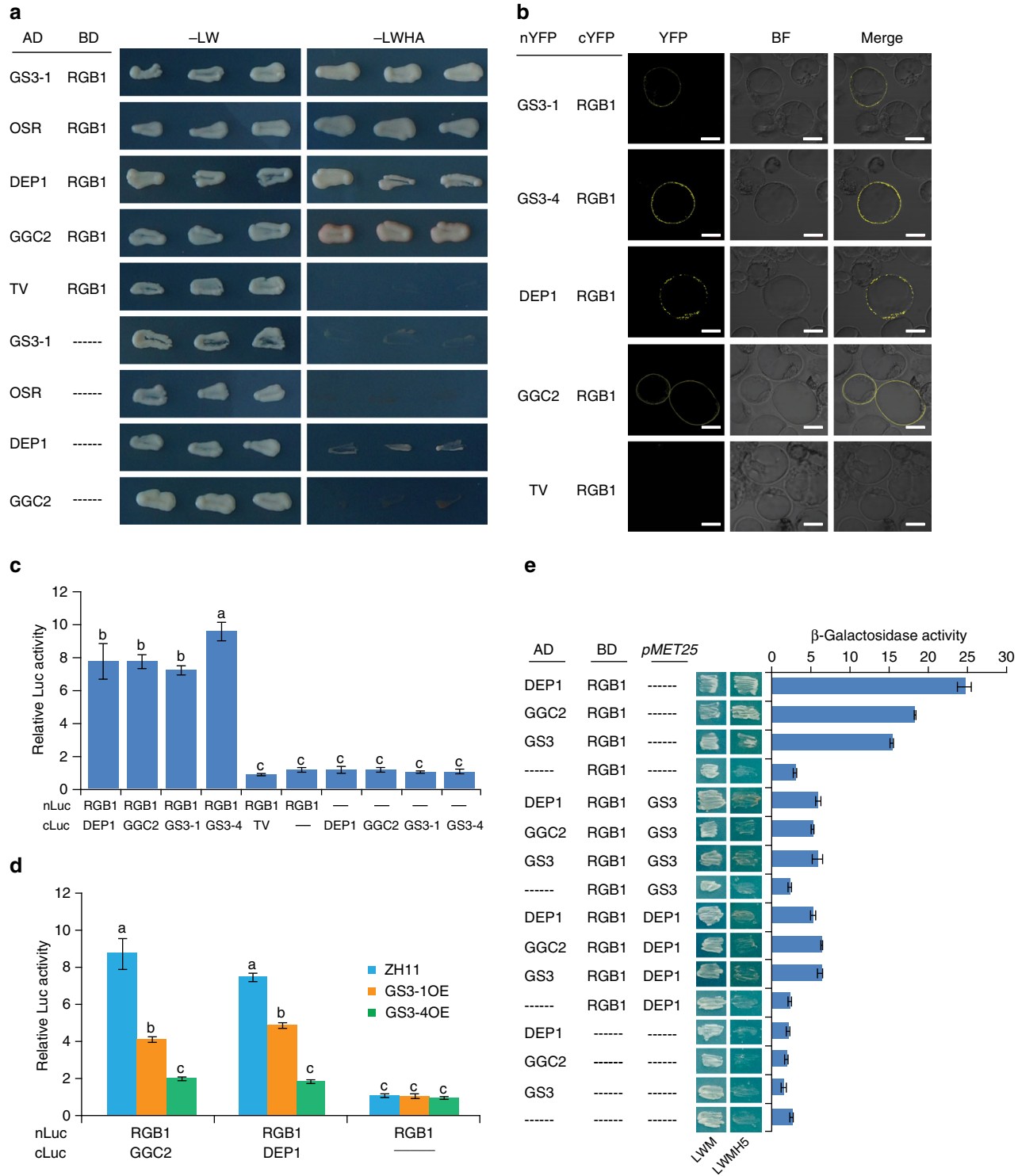

It should also be mentioned that similar behavior of DEP1 and GGC2 in grain size regulation does not mean that they have redundant functions at the whole plant level. In fact, these two genes show different expression profiles; constitutively over-expressing GGC2 (GGC2OE) resulted in shorter plants with fewer tillers and wider tillering angles than both wild-type and DEP1OE (Fig. 1c), suggesting that these two genes have different roles in growth and development.

A question thus arose: why did GS3 arise and how has it been maintained in the population at a predominantly high frequency? To answer this, we investigated the transgenic plants over-expressing DEP1 and GGC2, as well as RNAi of GS3 for yield related traits (Supplementary Table 1). Although these transgenes significantly increased grain length and weight, without exception, they reduced number of grains per panicle to the extents similar to their effects on grain length increase, indicating that the increased grain size is at the cost of fitness. Thus from fitness viewpoint, medium grain is probably the optimal for rice reproduction; increasing grain size to make it larger than medium would compromise fitness of the rice plant. As a strategy, the rice plant installed GS3 specifically to serve as a check to prevent the grain from becoming too large, as suggested by its specific expression in the developing panicle. Thus, while DEP1 and GGC2 function to promote grain size, which is essential for yield increase in breeding, GS3 plays a role to keep balance for maintaining fitness in response to natural selection.

Based on the understanding advanced in this study, the interactions of three Gγ proteins can be used for a predictable design of grain size in rice, by manipulating these genes, individually or in combination, to improve rice grain yield and quality. Moreover, as the Gγ proteins are highly conserved in a very wide range of plants, manipulating these proteins may provide a general strategy for modifying organ size and yield in crop breeding.

## Methods

**BLAST search and phylogenetic analyses.** Reciprocal BLASTP searches were done against the NCBI (http://blast.ncbi.nlm.nih.gov) and MSU (http://rice.plantbiology.msu.edu) databases, using the protein sequences of DEP1, GS3, and GGC2 (query cover >30%, E<e$^{-4}$, identity >30%). The phylogenetic tree was built based on the alignment of protein sequences, using MEGA 6.0.6[53]. The maximum-likelihood method was used with the parameters of Poisson model and bootstrap 1000 replicates.

**Field planting and trait measurement.** The rice plants were grown in the field in the normal rice growing seasons in Wuhan, China, and in Hainan in winter seasons. The planting density was 16.5 cm between plants in a row, and 26.5 cm between rows. Field management including irrigation, fertilizer application, and pest control followed essentially normal agricultural practices. Harvested grains were air-dried and stored at room temperature. Fully filled grains were used to measure the grain size using YTS system[54].

*Arabidopsis* plants were grown in soil at 22 °C under long-day conditions (16 h light/8 h dark). Petals and siliques were scanned to produce a digital image, and seeds were photographed under a Leica microscope. The petal length, silique length, and seed size were measured using image J software (https://imagej.nih.gov/ij/).

**Vector construction and plant transformation.** The coding sequences of GS3-1 and GS3-4 were amplified from full-length cDNA of Guangluai 4 (Osigcea013f09t3) in pBluescript SK2 (Stratagene). The coding sequences of DEP1 and dep1 were amplified from rice cultivar ZH11 by RT-PCR. The four sequences were inserted into pCAMBIA1301u to generate the constructs of GS3-1OE, GS3-4OE, DEP1OE, and dep1OE respectively, under an ubiquitin gene promoter. The coding sequence of GGC2 was also amplified from the rice cultivar ZH11 by RT-PCR, and was inserted to pCAMBIA1301s to generate the construct GGC2OE.

To construct the RNAi vectors for GS3, DEP1, and RGB1, ~500-bp fragment of the cDNA sequence was inserted into a pDS1301 vector modified from pCAMBIA1301[55].

The coding sequences of GS3-1 and dep1 were cloned into a pCAMBIA2300 vector under a 1.9-kb promoter of GS3 and a 2.9-kb promoter of DEP1 to generate the constructs GG, Gd, DG, and Dd, respectively. These vectors were transformed into gs3 (PFG_3A-03580), a homozygous mutant with a T-DNA insertion in the second intron of GS3, using *Agrobacterium tumefaciens* EHA105.

Genome sequences of AGG3[43] and GmDEP1 (from soybean variety Zhongdou 32), and cDNA sequence of ZmGS3[56] were amplified and inserted into pCAMBIA1301s to generate 35 S::AGG3, 35 S::GmDEP1, 35 S::ZmGS3, 35 S::AGG3$^{OSR}$, and 35 S::ZmGS3$^{OSR}$ vectors. All the constructs confirmed by sequencing were introduced into *Agrobacterium tumefaciens* EHA105 and transformed into ZH11 or DHX by *Agrobacterium*-mediated transformation as described in Lin and Zhang (2005) with minor modifications[57].

The DEP1, GGC2, and GS3 sequences were amplified and inserted into pCAMBIA1301s to construct 35 S::DEP1, 35 S::GGC2, and 35 S::GS3 vectors. They were introduced into an *Arabidopsis* agg3-2 mutant using *Agrobacterium tumefaciens* EHA105. The medium supplemented with hygromycin (30 mg ml$^{-1}$) was used to select the transformants. All primers for vector construction are listed in Supplementary Data 2.

**Generation of knock-out mutants using CRISPR/Cas9 technology.** The CRISPR/Cas9 system was used to generate the knock-out mutants[58]. The single mutants of DEP1$^{ko}$ (sgRNA designed in exon1) and GS3$^{ko}$ (sgRNA designed in exon1) were generated under the rice promoters of OsU3, OsU6b and OsU6a, OsU6c respectively. The double mutant GS3$^{ko}$DEP1$^{ko}$ was generated using the four sgRNAs in single mutants (Supplementary Fig. 3a–d).

The double mutant DEP1$^{ko}$GGC2$^{ko}$ was generated using the sgRNAs targeting both GGC2 and DEP1 in exon1, and the single mutant GGC2$^{ko}$ was generated using the sgRNAs targeting exon2. The triple mutant DEP1$^{ko}$GGC2$^{ko}$GS3-1$^{ko}$ was obtained using the sgRNAs targeting DEP1, GGC2, and GS3 in exon1 (Supplementary Fig. 3e–g)

The RGB1$^{ko}$ and RGA1$^{ko}$ mutants were generated using the three sgRNAs under the rice OsU3, OsU6a, and OsU6b promoters (Supplementary Fig. 4).

Mutations were confirmed by PCR sequencing in T$_0$ and T$_1$ generations.

**RNA extraction and qRT-PCR.** Tissues used for expression analysis included root, shoot, leaf sheath, flag leaf, panicles of 0.5 cm, 1 cm, 2 cm, 3.5 cm, 6.5 cm, 8 cm, 11 cm, 16.5 cm, 19 cm, 22 cm in length, and shoot apex at 10, 22, 34, 40, and 44 days after germination. Total RNA was isolated using an RNA extraction kit (TRIzol reagent; Invitrogen), and quantified using Nanodrop (Thermo). For qRT-PCR, approximately 3 μg of total RNA was reverse-transcribed using SuperScript reverse transcriptase (Invitrogen) in a volume of 20 μl to obtain the cDNA. qRT-PCR was carried out in a total volume of 25 μl containing 2 μl of the above reverse-transcribed product, 0.25 mM gene-specific primers,

**Fig. 5** Competition of the Gγ proteins in interacting with RGB1. **a** Interactions of GS3, DEP1 and GGC2 with RGB1 using yeast two-hybrid assay. GS3 interacts with RGB1 through the OSR domain but not the C-terminal cysteine-rich domain (TV). OSR: OSR domain of GS3, amino acid 1-94; TV: cysteine-rich domain of GS3, amino acid 95-231. AD: GAL4 activation domain; BD: GAL4 binding domain. -LW: selective medium lacking Trp and Leu; -LWHA: selective medium lacking Trp, Leu, His and Ade. **b** Interactions of GS3, DEP1 and GGC2 with RGB1 using BiFC assay. Bar = 20 μm. **c** Interactions of GS3, GGC2 and DEP1 (fused to C-terminal fragment of firefly luciferase) with RGB1 (fused to N-terminal fragment of firefly luciferase) using luciferase activity assay. Rice protoplasts with transient expression of RGB1-nLuc plus cLuc-DEP1, cLuc-GGC2, cLuc-GS3-1, and cLuc-GS3-4, but not the C-terminal cysteine-rich domain (TV), show high luciferase activity relative to the control. Empty vectors of cLuc plus RGB1-nLuc, and nLuc plus cLuc-DEP1, cLuc-GGC2, cLuc-GS3-1, and cLuc-GS3-4 were used as the negative control. Data are normalized to the internal control 35 S::REN. Values are given as mean ± SEM (n = 3). Different letters indicate significant differences ranked by the LSD test (P < 0.05). **d** Interactions of DEP1 and GGC2 with RGB1 in the background of ZH11, GS3-1OE, and GS3-4OE using luciferase activity assay. RGB1-nLuc plus cLuc-DEP1, and RGB1-nLuc plus cLuc-GGC2 are transiently expressed in the protoplasts of ZH11, GS3-1OE, and GS3-4OE, respectively. Empty vector of cLuc plus RGB1-nLuc is used as the negative control. Data are normalized to the internal control 35 S::REN. Values are given as mean ± SEM (n = 3). Different letters indicate significant differences ranked by the LSD test (P < 0.05). **e** Yeast three-hybrid assay for protein interactions of DEP1/GGC2 and GS3 with RGB1. The interaction of DEP1/GGC2 and GS3 with RGB1 is analyzed using fusions with AD (AD-DEP1, GGC2 or GS3) and BD (BD-RGB1). Empty vectors are used as a negative control. Quantitative analysis of interactions by β-galactosidase assay is shown. Data for all the assays are shown as mean ± SD (n = 3)

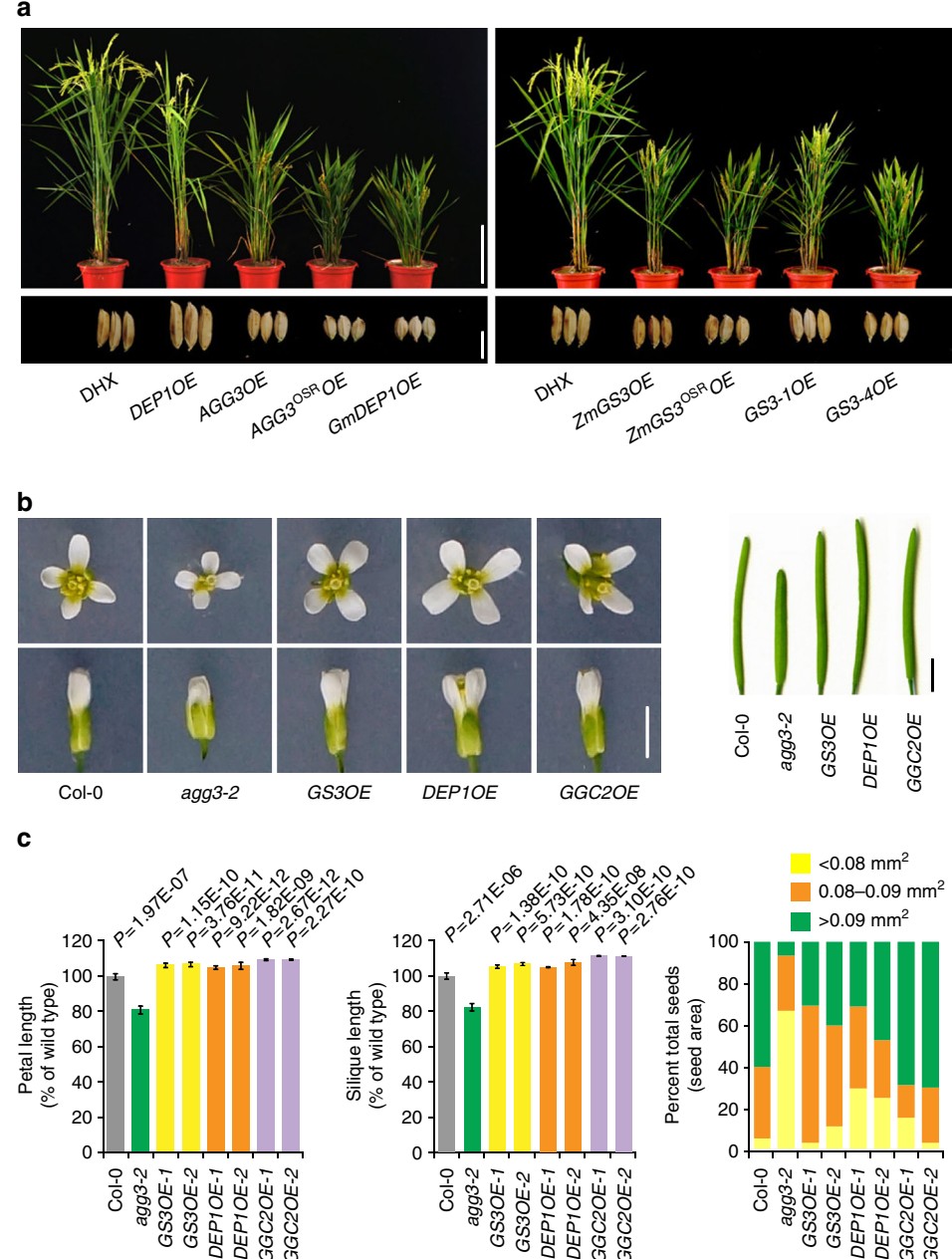

**Fig. 6** The functions of three Gγ proteins and their homologs in the genetic backgrounds of rice and *Arabidopsis*. **a** Whole plants and grains of DHX (wild type plants), *DEP1OE*, *AGG3OE*, *AGG3^OSR^OE*, *GmDEP1OE*, *ZmGS3OE*, *ZmGS3^OSR^OE*, *GS3-1OE*, and *GS3-4OE* rice plants in T$_1$ generation. Plant, bar = 20 cm; grains, bar = 5 mm. **b** Flowers and siliques of the wild type *Arabidopsis* strain Col-0, *agg3-2*, and *GS3OE*, *DEP1OE* and *GGC2OE* transgenes in *agg3-2* (a T-DNA mutant of *AGG3*) background. Bar = 2 mm. **c** Petal length, silique length, and seed area of the wild type Col-0, *agg3-2*, and *GS3OE*, *DEP1OE* and *GGC2OE* transgenes in *agg3-2* background. The fully opened flowers were measured for the sizes of petals. Values for petal length (*n* = 9) are given as mean ± SEM relative to the respective wild type values setting at 100%. Values for silique length of Col-0 (*n* = 12), *agg3-2* (*n* = 10), *GS3OE-1* (*n* = 16), *GS3OE-2* (*n* = 13), *DEP1OE-1* (*n* = 14), *DEP1OE-2* (*n* = 11), *GGC2OE*-1 (*n* = 10) and *GGC2OE*-2 (*n* = 10) are given as mean ± SEM relative to the respective wild type values setting at 100%. *P* values are given based on two-tailed *t*-tests. For seed size, the seeds are classified into three groups ( < 0.08, 0.08 to 0.09, and > 0.09 mm$^2$). Values of seed areas of Col-0 (*n* = 61), *agg3-2* (*n* = 147), *GS3OE-1* (*n* = 78), *GS3OE-2* (*n* = 74), *DEP1OE-1* (*n* = 96), *DEP1OE-2* (*n* = 90), *GGC2OE*-1 (*n* = 68) and *GGC2OE*-2 (*n* = 71) are expressed as a percentage of the total seed number analyzed

and 12.5 μl SYBR GreenMasterMix (Applied Biosystems) on an Applied Biosystems 7500 Real-time PCR system according to the manufacturer's instructions. Data were normalized with rice ubiquitin gene (LOC_Os03g13170). Primers used for qRT-PCR analysis are listed in Supplementary Data 2. The measurements were obtained using the relative quantification method[59].

**Yeast two-hybrid and yeast three-hybrid assays**. Yeast two-hybrid experiment was performed using the Matchmaker Two-Hybrid System (Clontech). The coding sequences of *RGB1*, *GS3*, *DEP1*, *GGC2* and their truncated forms were

amplified using primers listed in Supplementary Data 2. The obtained fragments were cloned into PGADT7 and PGBKT7 separately. The yeast two-hybrid assay was performed through cotransformation of the respective prey and bait vectors in the yeast (*Saccharomyces cerevisiae*) strain AH109 according to the lithium acetate transformation method[60]. The transformed yeast cells were selected by plating them onto synthetic dropout selection medium lacking Leu and Trp (SD/-LW). Interactions were assayed on synthetic dropout interaction medium lacking Leu, Trp, His and Ade (SD/-LWHA).

For the yeast three-hybrid assay[61], the prey vectors pGADT7-GS3, pGADT7-DEP1, and pGADT7-GGC2 were cotransformed with the different pBRIDGE (Clontech) bait vectors in the yeast strain AH109, respectively. Yeast strain AH109 was streaked three times to single colony on agar plates lacking Methionine (SD/-Met) since they tended to lose tolerance to this medium if not maintained on it routinely. Transformed cells were selected on synthetic dropout selection medium lacking Leu, Trp, Met (SD/-LWM) and interactions on SD/-Leu-Trp-Met-His (SD/-LWMH). The specificity of the stringency of the assay was tested by adding 5 mM 3-aminotriazole. For liquid β-galactosidase assay with CPRG as substrate, liquid cultures in SD-LWM were inoculated with three yeast colonies and incubated overnight. The calorimetric β-galactosidase assay of the supernatant and the following activity calculation were done, as described in the Clontech Yeast Protocols Handbook.

**BiFC assay**. The coding sequences of *GS3*, *DEP1*, *GGC2* and their variants were amplified using primers listed in Supplementary Data 2. The obtained fragments were cloned into pSPYNE(R)173 and pSPYCE(MR), respectively[62]. The plasmid mixtures were introduced into *Arabidopsis* protoplasts. After incubation in the dark overnight, the fluorescence was observed with Olympus FluoView FV1000.

**Luc activity assay**. The coding sequences of *GS3-1*, *GS3-4*, *DEP1* and *GGC2* were cloned to the C-terminal Luc fusion vector PUC19-cLUC, whereas *RGB1* was cloned to the N-terminal fusion Luc vector PUC19-nLUC[63]. The rice protoplast was isolated from rice shoot of 15-day-old seeding under dark conditions. For protein–protein interaction, N- and C-terminal *Luc* fusion genes were co-transformed with *35 S::REN* by the ratio 10:10:1, with the latter as the internal control. Luc and REN reporter activity were detected with a Dual-Luciferase Reporter Assay System (Promega E1910). Relative activity of the Luc reporter was expressed as the ratio of Luc to REN.

**Western blot**. Rice shoot (100 mg) was collected and lysed in 800 μl lysis buffer (50 mM Tris, pH 7.5, 150 mM NaCl, 0.1% Triton X-100, 0.2% Nonidet P-40, protease inhibitor cocktail (Roche, one tablet for 10 ml)) before centrifugation at 13,500 × g. for the supernatant. Plant extracts were loaded onto SDS–PAGE gel, and run until bromophenol blue was approximately touched the bottom of the gel. The proteins were transferred to a PVDF membrane. The membrane was blocked in 5% (wt/vol) nonfat milk for 1 h at room temperature and with anti-Flag antibody (Sigma Catalog F3165, dilution 1:10000) overnight at 4 °C. The membrane was washed in TBST (20 mM Tris, 150 mM NaCl, pH = 8.0 plus 0.05% Tween 20) four times, incubated with Goat anti-mouse secondary antibody (SouthernBiotech Catalog 1031-05, dilution 1:10000) for 1 h, and washed in TBST four times at room temperature. The uncropped scans with marker position are shown in Supplementary Fig. 6.

**Statistical analysis**. Statistical analysis was conducted using SPSS statistics 23.0 for Windows (IBM, Armonk, NY, USA). The two-tailed $t$ test was used for comparing agronomic traits of each transgenic line with the control. The Fisher's Least Significant Difference (LSD) test was used for multiple mean comparisons.

**Data availability**. All the vectors and the seeds of the plant materials generated in this study are available from the corresponding author upon request.

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

## Acknowledgements

We thank Dr. Yunhai Li for providing the *Arabidopsis* seeds, Drs. Jianbing Yan and Qing Li for the cDNA of *ZmGS3*, Dr. Wenhui Wei for the soybean seed, and Dr. Yaoguang Liu for the CRISPR construct. This work was supported by grants from the National Key Research and Development Program (2016YFD0100903), the National Natural Science Foundation of China (31771873), the Earmarked Fund for the China Agriculture Research System of China (CARS-01-05), and Bill & Melinda Gates Foundation.

## Author contributions

Q.Z. conceived and designed the research. S.S., Y.O., L.W. performed the experiments. H. M. and L.S. joined in vector construction. X.L. and J.X. contributed reagents/materials/ analysis tools. S.S., Y.O., and Q.Z. analyzed the data and wrote the manuscript. All the authors discussed the results and commented on the manuscript.

## Additional information

**Competing interests:** The authors declare no competing financial interests.

