## [Peer Review File · Nature Communications]

Reviewers' comments:

Reviewer #1 (Remarks to the Author):

In this manuscript, the authors systemically studied the genetic relationships among several subunits of G proteins involved in the grain length regulation in rice. The work is generally interesting but still requires further revision for publication in this journal.

Major points:

1. In DEP1OE/GS3-1OE hybrid transgenic plants, the interaction of GS3-RGB1 is stronger than that of DEP1-RGB1, and the plants show shorter grain. The protein level of DEP1 in DEP1OE/GS3-1OE and DEP1OE could be detected, and the result may explain that how GS3 and DEP1 antagonistically regulate grain size.
2. The C-terminal tail of GS3 is necessary for its degradation by the tail-mediated proteolysis pathway, which is crucial for grain size regulation. Because DEP1, GS3-4, and GS3-1 possess different tail types, another perspective that illustrates the difference between GS3 and DEP1 is thus presented if the C-terminal tail of DEP1 also mediates its degradation.
- 3 The genetic evidences showed that the functions of DEP1, GGC and GS3 are dependent on RGA1. However, this only indicates that the GS3, DEP1 and GGC2 cannot enhance grain length in RGA1-ko background. It is better to have more evidences, like the physical interaction.
4. Detection the protein level of RGB1 in GS3-1OE, GS3-4OE, DEP1OE, and GCC2OE transgenic plants using RGB1 antibody is preferred, in order to analyze the different influences of GS3, DEP1, and GCC2 on RGB1.
5. The competition of G γ proteins in interacting with RGB1 is critical to this work. However, based on current data, it is not very solid to see the functional competition, because the GS3-1 or GS3-4 was stably over-produced in transgenic plants. Moreover, the authors could try to perform the in vitro studies.
6. If the DEP1/dep1 protein was not found in DEP1F:Flag and dep1:Flag, how to confirm the

positive transgenic rice?

Minor points:

1. Line 59, “The OSR domain of GS3 showed 68.7% identity to DEP1 at the DNA level and 47% identity by protein sequence”. Usually the protein similarity is higher than nucleotide similarity, please confirm this.
2. It is interesting that, unlike DEP1, GGC2 largely reduces the plant height while increase grain length. The authors may need to explain and discuss this.
3. In many western blots (fig.4c,d), it looks like two bands (protein modification?). Please explain.
4. Some scale bars should be given in the figures (Fig.4, 6)

Reviewer #2 (Remarks to the Author):

Worked described by Sun and co-workers involves an impressive assembly of new genetic material to add to the well-studied field of grain size in rice. Publication of the results means that the corresponding authors agree to provide this material without delay or restriction to those researchers requesting this material. Without the ability for another researcher to verify the findings, then there is no value in its publication.

There is a large body of published knowledge dating back more than a decade showing that the $G\gamma$ subunits are important in rice grain size. In fact, the original rice mutant from Norman Borlaug’s Green Revolution of the 1940’s was d1 (daikoku 1), a dwarf rice resulting in loss of RGA1, the $G\alpha$ subunit in this pathway. Making a cereal dwarf reduces lodging and potentially increases yield dramatically by increasing harvest index. Unfortunately, mutations in the RGA1 confer many deleterious traits; the d1 mutant was never used for dwarfing. What are the agronomic traits of the $G\gamma$ gene mutations described here? Will pursuit of the $G\gamma$ subunit mutant traits suffer the fate of the d1 mutant?

The work here describes genetic epistasis between three members of the type C $G\gamma$ subunits (note that there are 2 other $G\gamma$ subunits not addressed here, likely because neither of these have been identified as involved in rice grain traits.) Figure 3 is the culmination of the epistasis data

set and it looks very similar to our state of knowledge in 2012 (see fig 2 of Botella JR 2012 TiPS).

Reviewer #3 (Remarks to the Author):

This paper is described the characterization of G protein subunits namely alpha, beta and three atypical gamma (GS3, DEP1 and GGC2) subunits, determining size in rice.

Generation of knock-out mutants, such as DEP1(KO), GGC2(KO), GS3(KO)DEP1(KO), DEP1(KO)GGC2(KO), GS3(KO)DEP1(KO)GGC2(KO) are original and important work. Reviewer agrees that DEP1 and GGC2 are positive regulators and GS3 is negative regulator in rice seed formation. The experiment of BiFC between beta and DEP1 in the GS3(OE) background is very informative. Reviewer, however, has some questions in this paper.

In the previous paper, RGA1 knock-out (d1) showed dwarfism and set small seeds. The mutant was not lethal. By RNAi method, severe suppression of RGB1 gene was proposed to be lethal. Mild suppression of RGB1 gene under d1 background caused more dwarfism (additive effect, not epistatic) than d1. From these observations, it is considered that RGB1 may have another independent function to RGA1, in addition to cooperation with RGA1 (Utsunomiya et al. Plant J. 2011 and Utsunomiya et al. Plant Signaling Behavior 2012). Authors showed that DEP1, GS3 and GGC2 were interacted with RGB1 by Y2H. Reviewer would like to ask the biochemical explain of the triple mutant GS3(KO)DEP1(KO)GGC2(KO) which phenotype is resemble to d1. How the triple mutant explains in the G protein signaling and the G protein cycling model containing G alpha, which is shown by Dr. Alan Jones? Does the author consider that the triple mutant can explain as an independent function to G alpha? Reviewer thinks that authors should describe a possible model occurred in the triple mutant, in the basis of G protein signaling.

Minor comments

- 1) In abstract, authors described that DEP1 and GGC2 increase grain length when in complex with G beta. Fig. 3 (a) model does not reflect this sentence. Authors should improve the model.
- 2) There are five Ggamma subunits in rice. Authors should explain RGG1 and RGG2 in introduction.
- 3) P7, lane 159: a functional dimer is better than a functional monomer
- 4) The result of double mutant, GS3(KO)DEP1(KO) should be add in Fig. 3 (b).

5) Western blot of Ubi::DEP1:Flag and Ubi::dep1:Flag in the presence of MG132 is necessary in Fig.4. When possible, panicle 1~3cm should be used for this WB.

Responses to Reviewers' Comments:

Reviewers' comments:

Reviewer #1 (Remarks to the Author):

In this manuscript, the authors systemically studied the genetic relationships among several subunits of G proteins involved in the grain length regulation in rice. The work is generally interesting but still requires further revision for publication in this journal.

Major points:

1. In DEP1OE/GS3-1OE hybrid transgenic plants, the interaction of GS3-RGB1 is stronger than that of DEP1-RGB1, and the plants show shorter grain. The protein level of DEP1 in DEP1OE/GS3-1OE and DEP1OE could be detected, and the result may explain that how GS3 and DEP1 antagonistically regulate grain size.

Answer: Many thanks for the valuable suggestion. Indeed *DEP1OE/GS3-1OE* hybrid showed phenotype of short grain, more similar to that of *GS3-1OE*. In the revised version we added a biochemical model (Fig. 3c) to explain the relation and phenotypic consequence of the competitive interactions of DEP1, GGC2 and GS3 with RGB1. We also added the result from yeast three hybrid assay supporting the model. We believe that the conclusion we made has a very solid base.

At the same time we have also been very interested in detecting the proteins with a more direct method, such as western-blot, as you suggested. Unfortunately, we have not been able to obtain an antibody for DEP1 or GS3, although we have tried very hard for many years. As an alternative, we generated transgenic plants expressing Flag tag-fused DEP1 and dep1. We showed that phenotypes of these transgenic plants were the same as the corresponding ones with elevated expression of *DEP1* and *dep1* without the Flag tag. But as indicated in the result, still the DEP1 and dep1 proteins could not be detected in DEP1:Flag and dep1:Flag transgenic plants. One possible reason is that the C-terminus of DEP1 might be cleaved in the post-translational processing (Taguchi-Shiobara et al, Breeding Science, 61:17-25 2011), leading to failure of detection of DEP1 using anti-flag antibody.

In addition, we investigated literatures on *DEP1*. No antibody for DEP1 was reported in previous studies, e.g. Huang et al. (Nature genetics, 41:494-497, 2009),

Zhou et al. (Genetics, 183:315-324, 2009), Sun et al. (Nature genetics, 46:652-656, 2014), and Wang et al. (Cell research, 27:1142-1156, 2017). After seeing a report from Zhang et al. (J Exp Bot, 66:6371-6384, 2015), we purchased antibody for DEP1 from Beijing Protein Innovation (Beijing, China). We saw protein band in *DEP1OE* and *dep1OE* plants using this antibody. However the fragment of an approximately 70-kDa in the western blot (Figure I) was not in agreement with the expected DEP1 protein size, although it seemed to match the gel photo provided by the Beijing Protein Innovation. We did not add this problematic data in the manuscript.

Figure I. Western blot analysis for *DEP1OE*, *DEP1OE/GS3-1OE*, and *DEP1OE/GS3-1Ri*. The left panel shows the western blot result from the manual provided by the Beijing Protein Innovation. M: marker.

2. The C-terminal tail of GS3 is necessary for its degradation by the tail-mediated proteolysis pathway, which is crucial for grain size regulation. Because DEP1, GS3-4, and GS3-1 possess different tail types, another perspective that illustrates the difference between GS3 and DEP1 is thus presented if the C-terminal tail of DEP1 also mediates its degradation.

Answer: Again I would use the model (Fig. 3c) to explain this point, which has now been stated as the first paragraph in the Discussion section: We infer that the long tails of DEP1 and GGC2 mediate signaling to promote cell proliferation thus increasing organ size, the function of the short tail of GS3 involves self-degradation, while the N-terminal OSR domain of these proteins binds to the G β protein (Fig. 3c). The net

outcome of the functional interactions among these genes together with their expression levels determines grain size.

3 The genetic evidences showed that the functions of DEP1, GGC2 and GS3 are dependent on RGA1. However, this only indicates that the GS3, DEP1 and GGC2 cannot enhance grain length in RGA1-ko background. It is better to have more evidences, like the physical interaction.

Answer: Many thanks for the valuable suggestion. We had investigated the physical interaction of GS3, DEP1 and GGC2 with RGA using yeast two-hybrid and luciferase activity assays. However, we did not detect interaction of GS3, DEP1 or GGC2 with RGA1 using both of these two approaches (Figure II). Evidence from literature indicates that plant G proteins are self-activating (Daisuke Urano and Alan M. Jones, *Annu Rev Plant Biol*, 65:8.1-8.20, 2014), which implies that G α and G γ may never be in physical contact. Our results seemed to be consistent with such evidence.

Figure II. Assays of interaction between G γ proteins and RGA1.

(a) Interactions of GS3, DEP1 and GGC2 with RGA1 using Yeast two-hybrid assay. TUD1 that interacted with RGA1 is used as a positive control. (b) Interactions of GS3, GGC2 and DEP1 (fused to C-terminal fragment of firefly luciferase) with RGB1 and RGA1 (fused to N-terminal fragment of firefly luciferase) using luciferase activity assay. Rice protoplasts with transient expression of RGB1-nLuc plus cLuc-DEP1, cLuc-GGC2, cLuc-GS3-1, and cLuc-GS3-4 show high luciferase activity relative to the control with empty vector. TV: cysteine-rich domain of GS3, amino acid 95-231. No luciferase activity is detected in rice protoplasts with transient expression of

RGA1-nLuc plus cLuc-DEP1, cLuc-GGC2, cLuc-GS3-1, and cLuc-GS3-4. Empty vectors of cLuc plus RGA1-nLuc, and nLuc plus cLuc-DEP1, cLuc-GGC2, cLuc-GS3-1, and cLuc-GS3-4 are used as the negative control. Data are normalized to the internal control 35S::REN. Values are given as mean \pm SEM (n = 3).

4. Detection the protein level of RGB1 in GS3-1OE, GS3-4OE, DEP1OE, and GCC2OE transgenic plants using RGB1 antibody is preferred, in order to analyze the different influences of GS3, DEP1, and GCC2 on RGB1.

Answer: Many thanks for the valid suggestion. Again because of difficulty in acquiring an antibody of RGB1, we investigated the expression levels of *RGA1* and *RGB1* in *GS3-1OE*, *GS3-4OE*, *DEP1OE*, and *GCC2OE* transgenic plants (Figure III). Both *RGA1* and *RGB1* showed similar expression levels across these overexpression plants, suggesting that the protein levels of RGB1 and RGA1 may not have been affected much by the transgenes.

Figure III. Relative expression levels of *RGA1* and *RGB1* in the transgenic plants of *GS3-1OE*, *GS3-4OE*, *DEP1OE*, and *GCC2OE*. WT, the wild type Zhonghua 11. Transformants from two independent T₂ homozygous lines were assayed for each transgene. Rice *ubiquitin* is used as the reference gene. Data for all the assays are shown as mean \pm SEM (n = 3).

5. The competition of Gy proteins in interacting with RGB1 is critical to this work. However, based on current data, it is not very solid to see the functional competition,

because the GS3-1 or GS3-4 was stably over-produced in transgenic plants. Moreover, the authors could try to perform the in vitro studies.

Answer: In the revised version we added a biochemical model (Fig. 3c) to explain the relation and phenotypic consequence of the competitive interactions of DEP1, GGC2 and GS3 with RGB1. We also performed a yeast three hybrid assay for supporting the model. The competition of GS3, DEP1 and GGC2 in interaction with RGB1 was confirmed by the yeast three-hybrid assay, such that the interaction between DEP1/GGC2 and RGB1 was disrupted by the expression of GS3, and the reverse is also the case (Fig. 5e). We believe that the conclusion we made has a very solid base.

Figure 5e. Yeast three-hybrid assay for protein interactions of DEP1/GGC2 and GS3 with RGB1. The interaction of DEP1/GGC2 and GS3 with RGB1 is analyzed using fusions with the GAL4 activation domain (AD-DEP1, GGC2 or GS3) and the binding domain (BD-RGB1). Empty vectors are used as a negative control. Quantitative analysis of interactions by β -galactosidase assay is shown. Data for all the assays are shown as mean \pm SD (n = 3).

6. If the DEP1/dep1 protein was not found in DEP1F:Flag and dep1:Flag, how to confirm the positive transgenic rice?

Answer: We usually perform co-segregation analysis in T₁ families for the genetic effect of transgene. We showed that the phenotype co-segregated with the transformed *DEP1/dep1* (Figure IV). We also detected increased expression level of *DEP1/dep1* in the positive plants (Supplementary Fig. 1).

Figure IV. Co-segregation analysis of the *dep1OE* (a) and *DEP1OE* (b) transgenic plants between the genotype and grain length. The blue indicates the transgenic

positive plants, and the red ones show the negative transgenes.

Minor points:

1. Line 59, “The OSR domain of GS3 showed 68.7% identity to DEP1 at the DNA level and 47% identity by protein sequence”. Usually the protein similarity is higher than nucleotide similarity, please confirm this.

Answer: The OSR domain of GS3 protein showed 47% identity to DEP1 using BLASTP search in NCBI database, whereas no *DEP1* hit was identified in NCBI using BLASTN search.

Table. BLASTP result using GS3-OSR protein as a query in NCBI database.

Select for downloading or viewing reports	Description	Max score	Total score	Query cover	E value	Ident	Accession
Select seq	PREDICTED:						
ref XP_015610892.1 (DEP1)	keratin-associated protein 5-5 [Oryza sativa Japonica Group]	67.4	67.4	95%	6e-11	47%	XP_015610892.1

Besides, the DNA sequence of *GS3-OSR* showed 68.7% identity to *DEP1* using BLASTN search in MSU database. And the protein sequence of *GS3-OSR* showed 50% identity to *DEP1* using BLASTP search in this database.

Table. BLASTN result using *GS3-OSR* sequence as a query in MSU database.

Acc	Description	Hit Score	E value	Num HSPs	Top Query Cov	Top Id
LOC_Os09g26999 (DEP1)	genomic keratin-associated protein 5-4, putative, expressed	245	7.8e-05	1	46.18%	68.70%
LOC_Os03g42710	genomic WD-40 repeat family protein, putative, expressed	220	0.00089	1	37.75%	71.58%
LOC_Os12g35580	genomic riboflavin synthase alpha chain, putative, expressed	220	0.00095	1	52.21%	65.67%
LOC_Os03g47350	genomic expressed protein	203	0.0011	1	38.96%	72.82%

Table. BLASTP result using GS3-OSR protein as a query in MSU database.

Acc	Description	Hit Score	E value	Num HSPs	Top Query Cov	Top Id
LOC_Os09g26999.1 (DEP1)	protein keratin-associated protein 5-4, putative, expressed	186	1.4e-14	1	87.06%	50.00%
LOC_Os09g26999.2	protein keratin-associated protein 5-4, putative, expressed	146	4.5e-11	1	62.35%	54.55%

Acc	Description	Hit Score	E value	Num HSPs	Top Query Cov	Top Id
LOC_Os02g04520.1	protein AGG2, putative, expressed	85	0.00014	1	69.41%	34.43%
LOC_Os01g52670.1	protein expressed protein	71	0.011	1	34.12%	48.28%

We are confident about the results of BLAST search. We used both the BLASTN and BLASTP data from MSU database in the revised manuscript for a more consistent result.

2. It is interesting that, unlike DEP1, GGC2 largely reduces the plant height while increase grain length. The authors may need to explain and discuss this.

Answer: Thanks for raising such an important point. We rewrote the first paragraph of Discussion to address this.

3. In many western blots (fig.4c,d), it looks like two bands (protein modification?). Please explain.

Answer: Many thanks for pointing it out. The two bands of GS3-1 protein were likely due to protein modification of ubiquitination and subsequent degradation. We explained it in the figure legend.

4. Some scale bars should be given in the figures (Fig.4, 6)

Answer: Many thanks for pointing it out. We added the scale bars for the plants.

Reviewer #2 (Remarks to the Author):

Worked described by Sun and co-workers involves an impressive assembly of new genetic material to add to the well-studied field of grain size in rice. Publication of the results means that the corresponding authors agree to provide this material without delay or restriction to those researchers requesting this material. Without the ability for another researcher to verify the findings, then there is no value in its publication.

Answer: Many thanks for your positive comments. We are happy to provide the materials to researchers in need. Actually, we have already provided many rice

materials for *GS3* (transgenic plants/seeds generated in Mao et al, PNAS, 107: 19579-19584, 2010) to a number of scientists including research groups of Alan M. Jones and José Ramón Botella.

There is a large body of published knowledge dating back more than a decade showing that the $G\gamma$ subunits are important in rice grain size. In fact, the original rice mutant from Norman Borlaug's Green Revolution of the 1940's was d1 (daikoku 1), a dwarf rice resulting in loss of RGA1, the $G\alpha$ subunit in this pathway. Making a cereal dwarf reduces lodging and potentially increases yield dramatically by increasing harvest index. Unfortunately, mutations in the RGA1 confer many deleterious traits; the d1 mutant was never used for dwarfing. What are the agronomic traits of the $G\gamma$ gene mutations described here? Will pursuit of the $G\gamma$ subunit mutant traits suffer the fate of the d1 mutant?

Answer: Thanks for reviewing such a history, and this is a valid concern. Actually, Jose Botella wrote a review article in 2012 published in TIPS, asking: Can heterotrimeric G proteins help to feed the world? He already offered a very promising perspective, which we quite agree and also feel encouraged.

In fact, *GS3* is the most important regulator of grain length, which has contributed to rice improvement of both yield and quality in a global scale. My own estimate is that at least 50% of *indica* rice varieties worldwide have the long grain allele of *GS3*, and many of the *japonica* cultivars especially those of tropical *japonica* varieties in the Americas also incorporated the long grain allele. The *DEP1* gene has also found important use in rice breeding. For example the *dep1* allele has been used for modifying panicle architecture resulting in elite varieties such as Wuyungeng, Shennong 265, Liaojing 5, Qianchonglang, and their derivatives in China. The *dep1* allele has also found use in improving nitrogen use efficiency (Sun et al 2014 Nat Genet). Not much is known of *GGC2* at present providing a target for future work.

Overall, we are quite confident that the pursuit of these $G\gamma$ subunits would be useful in rice improvement.

The work here describes genetic epistasis between three members of the type C G γ subunits (note that there are 2 other G γ subunits not addressed here, likely because neither of these have been identified as involved in rice grain traits).

Answer: We added the description of RGG1 and RGG2 in introduction.

Figure 3 is the culmination of the epistasis data set and it looks very similar to our state of knowledge in 2012 (see fig 2 of Botella JR 2012 TiPS).

Answer: The paper of Botella (2012) should be gratefully acknowledged in that it spelt out that both GS3 and DEP1 are G \$\gamma\$ proteins, which has been heuristic to us, suggesting that these genes should be considered together in understanding the mechanism. So this is what we have done in this work. The most important advance of this work, however, is the interactions AMONG the three genes in grain size regulation, rather than interactions within individual genes as groped in Botella (2012) and also the paper by Mao et al (2010). We added a biochemical mode to highlight this finding (Fig. 3c).

Reviewer #3 (Remarks to the Author):

This paper is described the characterization of G protein subunits namely alpha, beta and three atypical gamma (GS3, DEP1 and GGC2) subunits, determining size in rice.

Generation of knock-out mutants, such as DEP1(KO), GGC2(KO), GS3(KO)DEP1(KO), DEP1(KO)GGC2(KO), GS3(KO)DEP1(KO)GGC2(KO) are original and important work. Reviewer agrees that DEP1 and GGC2 are positive regulators and GS3 is negative regulator in rice seed formation. The experiment of BiFC between beta and DEP1 in the GS3(OE) background is very informative. Reviewer, however, has some questions in this paper.

In the previous paper, RGA1 knock-out (d1) showed dwarfism and set small seeds. The mutant was not lethal. By RNAi method, severe suppression of RGB1 gene was proposed to be lethal. Mild suppression of RGB1 gene under d1 background caused more dwarfism (additive effect, not epistatic) than d1. From these observations, it is considered that RGB1 may have another independent function to RGA1, in

addition to cooperation with RGA1 (Utsunomiya et al. Plant J. 2011 and Utsunomiya et al. Plant Signaling Behavior 2012). Authors showed that DEP1, GS3 and GGC2 were interacted with RGB1 by Y2H. Reviewer would like to ask the biochemical explain of the triple mutant GS3(KO)DEP1(KO)GGC2(KO) which phenotype is resemble to d1. How the triple mutant explains in the G protein signaling and the G protein cycling model containing G alpha, which is shown by Dr. Alan Jones? Does the author consider that the triple mutant can explain as an independent function to G alpha? Reviewer thinks that authors should descript a possible model occurred in the triple mutant, in the basis of G protein signaling.

Answer: Many thanks for your positive comments and great suggestion. We have proposed a biochemical model to explain how these G γ proteins work (Fig. 3c). In this model, DEP1 and GGC2, when coupled with RGB1, promote grain size by tail-mediated signaling. GS3, though having no function in promoting or inhibiting grain size, reduces grain size by blocking the interaction of DEP1 and GGC2 with RGB1. The tail-mediated self-degradation of GS3 in the RGB1-GS3 complex does not severely block the DEP1/GGC2-RGB1 interactions, thus a plant carrying *GS3-1* produces medium grain. Without the tail (GS3-4), accumulated GS3-4 would largely occupy RGB1, thus producing short grain. In *DEP1^{ko}GGC2^{ko}GS3-1^{ko}* triple mutant, the *DEP1/GGC2* tail mediated signaling was no longer functional, thus would produce very small grain, which might be very similar to the phenotype of d1 mutant lacking RGA1, as you suggested.

We agree that RGB1, when activated, may function independently of RGA1. We thus put RGB1 on the very basic level of our genetic model, indicating that it plays crucial roles in plant survival and growth, in addition to grain size regulation.

Minor comments

1) In abstract, authors described that DEP1 and GGC2 increase grain length when in complex with G beta. Fig. 3 (a) model does not reflect this sentence. Authors should improve the model.

Answer: We have added a figure to explain how these G γ proteins work (Fig. 3c).

2) There are five G gamma subunits in rice. Authors should explain RGG1 and RGG2 in introduction.

Answer: We added the description of RGG1 and RGG2 in introduction.

3) P7, lane 159: a functional dimer is better than a functional monomer.

Answer: Many thanks for pointing it out. We modified this as suggested.

4) The result of double mutant, GS3(KO)DEP1(KO) should be add in Fig. 3 (b).

Answer: We revised the figure as suggested.

5) Western blot of Ubi::DEP1:Flag and Ubi::dep1:Flag in the presence of MG132 is necessary in Fig.4. When possible, panicle 1~3cm should be used for this WB.

Answer: Thanks for raising this issue. Actually this is a problem that has bothered us very long. Although we have tried our best, we have not been able to obtain an antibody for DEP1. As an alternative, we generated transgenic plants expressing Flag tag-fused DEP1 and dep1. Phenotypes of these transgenic plants were the same as the corresponding ones with elevated expression of *DEP1* and *dep1* without the Flag tag. But as indicated in the result, the DEP1 and dep1 proteins could not be detected in DEP1:Flag and dep1:Flag transgenic plants. One possible reason is that the C-terminus of DEP1 might be cleaved in the post-translational processing (Taguchi-Shiobara et al, Breeding Science, 61:17-25 2011), leading to failure of detection of DEP1 using anti-flag antibody.

In addition, we investigated literatures on *DEP1*. No antibody for DEP1 was reported in previous studies, e.g. Huang et al. (Nature genetics, 41:494-497, 2009), Zhou et al. (Genetics, 183:315-324, 2009), Sun et al. (Nature genetics, 46:652-656, 2014), and Wang et al. (Cell research, 27:1142-1156, 2017). Based on report from Zhang et al. (J Exp Bot, 66:6371-6384, 2015), we purchased antibody for DEP1 from Beijing Protein Innovation (Beijing, China). We saw protein band in *DEP1OE* and *dep1OE* plants using this antibody. However the fragment of an approximately

70-kDa in the western blot (Figure V) was not in agreement with the expected DEP1 protein size, although it conformed to the gel photo provided by the Beijing Protein Innovation. We did not add this problematic data in the manuscript.

Figure V. Western blot analysis for *DEP1OE* and *dep1OE*. The left panel shows the western blot result from the manual provided by the Beijing Protein Innovation. M: marker.

Reviewers' comments:

Reviewer #1 (Remarks to the Author):

Most of the concerns have been addressed in the revised manuscript. There are only minor suggestions:

1. Please try to polish the English to be more refined and professional throughout the text.
2. The relationship between G protein and grain size regulation is recommended to be more introduced in the Introduction section.

Reviewer #3 (Remarks to the Author):

Authors have addressed my comments and have improved the paper. Reviewer thinks that this paper is an important contribution to the plant G protein signaling field.

Minor points:

Authors should add the result of DEP1OE/GS3OE which set short grain in Figure 3 (b).

Authors should explain numbers, 6, 12, and 20 in Figure 4 (d).

Reviewer #4 (Remarks to the Author):

The manuscript “A G-protein pathway determines grain size in rice” by Sun et al., is an attempt to answer an incredibly important question in the plant G-protein signaling and find the genetic/mechanistic basis of a key phenotype. Grain size determination is inarguably an important trait for cereal crops. Ever since the discovery of GS3 allele as a major QTL for grain size in rice, it has been an important area of research with an aim to characterize it in exquisite details, which would potentially help generate “on-demand sized grains”. This is also the premise of the current manuscript. However, after reading this manuscript over and over, I am left with more questions than answers. Overall, the authors present a model that is expected to explain their findings, but it does not. Here are my major concerns:

Introduction:

It is not comprehensive. The authors have left out certain important papers for example, a long field study in barley has concluded that the role of these proteins is highly dependent on genetic background and environmental conditions (Wendt et al., 2016); the role of DEP1 as a major QTL

for Nitrogen use efficiency (Sun et al., 2014; this paper is also important for a potential interaction between DEP1 and Ga protein). The paper on detailed characterization of Gg proteins from soybean, overexpression of AGG3 protein in Camelina (Roy Choudhury et al., 2011, 2014), a paper by Trusov et al, describing all known Gg proteins in multiple plant species (2012). There are many typos for example, the use of word 'statue' to describe 'stature', 'sergeants' for segregant, 'dimer' as 'dimmer' and so on.

The authors state that 'plant genomes contain much fewer G proteins' (line 41). The soybean genome codes for 4 canonical and 10 extra-large Ga, 4 Gb and 12 Gg proteins, in addition to potential variants. How is this fewer than the reported human numbers?

The authors state that 'biological functions of plant G proteins have been much less studied (line 42); again incorrect. Even though the mechanisms are not known, the function of plant G-proteins is extremely well-known and extensively characterized.

It has now been well-established the XLG proteins are a part of G-protein heterotrimer in plants. The authors do not pay any attention to this either while introducing G-proteins, or while presenting the model. This is hugely consequential in the context of G-protein signaling. The GS3, DEP1 type Gg proteins are as different from canonical Gg as the XLGs from canonical Ga.

Methods:

A lot of experiments have been performed on T1 lines, which I see as a huge problem. The field is complicated enough and adding more data from lines which are potentially segregating is a problem. This becomes even more complicated when they are using a particular line to express another gene or silence another gene. I did not find any details of proper genetic characterization of these lines (copy number, homozygosity, etc?). The authors' group is extremely well-equipped to these things and they should really use homozygous, T3 lines to do the experiments. I do not believe that yeast 2 or 3 hybrid assays are sufficient to determine the strength of interaction between interacting proteins. It is a highly heterologous system and is prone to way too many artefacts. In addition, there are two more things that I would like to point out:

- 1) In both rice and Arabidopsis published reports suggest an interaction between Ga and Gg proteins. The authors do not see that. The rice paper in fact also provides genetic evidence to support their findings. So what is the authors' response? Why this discrepancy?
- 2) In soybean the type III Gg proteins have been shown to interact strongly with each other, which is not surprising given the presence of an extremely Cysteine rich region, which might just make them 'sticky'. If such a thing happens with the rice proteins as well, it makes the interpretation of data in Fig. 5 even more difficult.

Results:

The overall results are explained in the context of the model presented in Fig. 3. However, the model loses validity as the XLG proteins are not included in it. XLG proteins work with Gb proteins and can't be left out of the equation, if the authors are trying to explain the mechanism based on G-protein signaling. Maybe the phenotypes of Gb mutant are stronger than the Ga mutant because one Gb is shared between three XLGs and one Ga in plants such as rice.

The experimental evidence in support of tail mediated self-degradation is weak.

The authors suggest that GS3 by itself has no role in regulating grain size, its regulation is only via affecting the DEP1 and GCG2. However, given the important role different domains play it cannot be ascertained with the data the authors have. There may be an epistatic relationship between different genes. These results are especially difficult to be conclusive as the genetic analysis of different mutants is not complete at this stage.

One of the GS3 alleles, Chuan3, has been reported to have extremely small-sized grain. How does the grain size of double or triple mutants of Gg or RGA mutants compare to that? Even if they are in different genetic background, a size ratio between WT and mutant allele can be compared.

Because the regulation of the phenotypes seems to be so strongly dependent on the site of the mutation and the presence of potential peptide, how is one supposed to interpret the results of RNAi mediated suppression?

There are a lot of other issues and I can go on, but the bottom line is, I don't think the results presented in the manuscript justify the conclusions drawn. The authors have generated a HUGE amount of genetic material and have performed a LOT of experiments, but it is not uncovering the mechanism of regulation by these proteins. This is an important mechanism to explore, and it has the potential to positively affect food security, but a lot more is needed. May be experiments such as using different tail lengths of a protein expressed in a true null background, making chimeric proteins (OSR of DEP1 with tail of GS3 etc.) to systematically elucidate the roles of different domains and different tail length will be more useful.

Responses to Reviewers' Comments:

Reviewers' comments:

Reviewer #1 (Remarks to the Author): Most of the concerns have been addressed in the revised manuscript. There are only minor suggestions:

1. Please try to polish the English to be more refined and professional throughout the text.

Answer: Thanks for your suggestion. We have done more rounds of editing.

2. The relationship between G protein and grain size regulation is recommended to be more introduced in the Introduction section.

Answer: Thanks for your suggestion. We modified the Introduction.

Reviewer #3 (Remarks to the Author):

Authors have addressed my comments and have improved the paper. Reviewer thinks that this paper is an important contribution to the plant G protein signaling field.

Minor points: Authors should add the result of DEP1OE/GS3OE which set short grain in Figure 3 (b).

Answer: Thanks. We have revised Figure 3b as suggested.

Authors should explain numbers, 6, 12, and 20 in Figure 4 (d).

Answer: We have explained the numbers in the figure legend.

Reviewer #4 (Remarks to the Author):

The manuscript “A G-protein pathway determines grain size in rice” by Sun et al., is an attempt to answer an incredibly important question in the plant G-protein signaling and find the genetic/mechanistic basis of a key phenotype. Grain size determination is inarguably an important trait for cereal crops. Ever since the discovery of GS3 allele as a major QTL for grain size in rice, it has been an important area of research with an aim to characterize it in exquisite details, which would potentially help generate “on-demand sized grains”. This is also the premise of the current manuscript. However, after reading this manuscript over and over, I am left with more questions than answers. Overall, the authors present a model that is expected to explain their findings, but it does not. Here are my major concerns:

Introduction:

It is not comprehensive. The authors have left out certain important papers for example, a long field study in barley has concluded that the role of these proteins is highly dependent on genetic background and environmental conditions (Wendt et al., 2016); the role of DEP1 as a major QTL for Nitrogen use efficiency (Sun et al., 2014; this paper is also important for a potential interaction between DEP1 and Ga protein). The paper on detailed characterization of Gg proteins from soybean, overexpression of AGG3 protein in Camelina (Roy Choudhury et al., 2011, 2014), a paper by Trusov et al, describing all known Gg proteins in multiple plant species (2012).

Answer: We have added two references concerning DEP1 (Sun et al., 2014) and diversity of plant Gγs (Trusov et al., 2012) in the introduction section. We also added a couple of references describing G proteins in *Arabidopsis*. However, because of length constraint, we did not add the other references (Wendt et al., 2016, PLoS ONE; Choudhury et al., 2011, PLoS ONE; Choudhury et al., 2014, PBJ) in the introduction section, as they are not very relevant to our work.

There are many typos for example, the use of word ‘statue’ to describe ‘stature’, ‘sergeants’ for segregant, ‘dimer’ as ‘dimmer’ and so on.

Answer: Thanks for pointing out. We corrected the typos.

The authors state that ‘plant genomes contain much fewer G proteins’ (line 41). The soybean genome codes for 4 canonical and 10 extra-large Ga, 4 Gb and 12 Gg proteins, in addition to potential variants. How is this fewer than the reported human numbers? The authors state that ‘biological functions of plant G proteins have been much less studied (line 42); again incorrect. Even though the mechanisms are not known, the function of plant G-proteins is extremely well-known and extensively characterized.

Answer: The statement of “plant genomes contain much fewer G proteins” gives an overview of the G proteins in plants and comparing plants and animal in general rather than a specific species, certainly not all the plants against human. To avoid ambiguity, we revised this sentence to “Most plant genomes contain much fewer G proteins (Daisuke Urano and Alan M. Jones, Annu Rev Plant Biol, 65:8.1-8.20, 2014)”.

The biological functions of plant G proteins are surely much less studied than in animals.

It has now been well-established the XLG proteins are a part of G-protein heterotrimer in plants. The authors do not pay any attention to this either while introducing G-proteins, or while presenting the model. This is hugely consequential in the context of G-protein signaling. The GS3, DEP1 type Gg proteins are as different from canonical Gg as the XLGs from canonical Ga.

Answer: XLG proteins were involved in disease resistance in *Arabidopsis*. This is not relevant to our work as these proteins have not been studied in rice, neither do they have relationship with grain size regulation, and thus are not the focus of the present work.

Methods:

A lot of experiments have been performed on T1 lines, which I see as a huge problem. The field is complicated enough and adding more data from lines which are potentially segregating is a problem. This becomes even more complicated when they are using a particular line to express another gene or silence another gene. I did not find any details of proper genetic characterization of these lines (copy number, homozygosity, etc?). The authors' group is extremely well-equipped to these things and they should really use homozygous, T3 lines to do the experiments.

Answer: All of our genetic analyses were performed using co-segregation analyses in T₁ generation, which is a "golden standard" set by the rice genetics community in the 1990s for transformation study. In our manuscript, we presented data from at least two independent T₁ transgenic families, and portions of the results were even further confirmed by T₂ families. This is a much more rigorous analysis than an analysis based on comparison between homozygous transgenic lines against wild type, because of the somaclonal variations in the genetic backgrounds resulting from tissue culture that were ubiquitously detected in rice transgenic plants. We believe that this type of analysis should also be a standard for any transformation involving tissue culture.

I do not believe that yeast 2 or 3 hybrid assays are sufficient to determine the strength of interaction between interacting proteins. It is a highly heterologous system and is prone to way too many artefacts.

Answer: The interactions between these proteins were determined by combinatorial approaches including Yeast two-hybrid and yeast three-hybrid assays, BiFC assay, Luc activity assay, as well as genetic analyses. Like all the papers published so far, *in vitro* experiments provide clues and supportive data. It is somewhat up to the readers to judge the quality of the work.

In addition, there are two more things that I would like to point out:

1) In both rice and Arabidopsis published reports suggest an interaction between Ga and Gg proteins. The authors do not see that. The rice paper in fact also provides genetic evidence to support their findings. So what is the authors' response? Why this discrepancy?

Answer: We had investigated the physical interaction of GS3, DEP1 and GGC2 with RGA using yeast two-hybrid and luciferase activity assays, and repeated this experiments many times in several years. However, we did not detect interaction of GS3, DEP1 or GGC2 with RGA1 using both of these two approaches (see Figure below). Evidence from literature indicates that plant G proteins are self-activating (Daisuke Urano and Alan M. Jones, *Annu Rev Plant Biol*, 65:8.1-8.20, 2014), which implies that G α and G γ may never be in physical contact. If that is the case, there may be no reason to expect physical interaction between them. Reviewer 1 also raised this issue before and he is now satisfied with our answer. We have discussed with G protein experts in other countries about the possible interactions between G α and G γ , but have not obtained a really positive answer.

Regarding the “rice work” mentioned by this reviewer, the G α -G γ interaction in Sun et al. was only performed by BiFC analysis in tobacco leaves. This result was probably not solid enough without support from other *in vivo* and *in vitro* evidences. Frankly we have some doubt about that result.

Figure. Assays of interaction between G γ proteins and RGA1.

(a) Interactions of GS3, DEP1 and GGC2 with RGA1 using Yeast two-hybrid assay. TUD1 that interacted with RGA1 is used as a positive control. (b) Interactions of GS3, GGC2 and DEP1 (fused to C-terminal fragment of firefly luciferase) with RGB1 and RGA1 (fused to N-terminal fragment of firefly luciferase) using luciferase activity assay. Rice protoplasts with transient expression of RGB1-nLuc plus cLuc-DEP1, cLuc-GGC2, cLuc-GS3-1, and cLuc-GS3-4 show high luciferase activity relative to the control with empty vector. TV: cysteine-rich domain of GS3, amino acid 95-231. No luciferase activity is detected in rice protoplasts with transient expression of RGA1-nLuc plus cLuc-DEP1, cLuc-GGC2, cLuc-GS3-1, and cLuc-GS3-4. Empty vectors of cLuc plus RGA1-nLuc, and nLuc plus cLuc-DEP1, cLuc-GGC2, cLuc-GS3-1, and cLuc-GS3-4 are used as the negative control. Data are normalized to the internal control 35S::REN. Values are given as mean \pm SEM (n = 3).

2) In soybean the type III G γ proteins have been shown to interact strongly with each other, which is not surprising given the presence of an extremely Cysteine rich region, which might just make them 'sticky'. If such a thing happens with the rice proteins as well, it makes the interpretation of data in Fig. 5 even more difficult.

Answer: The G γ proteins are diverged between dicot and monocot, as we showed in our result. Thus we proposed our model based on rice result, but not from soybean.

Results:

The overall results are explained in the context of the model presented in Fig. 3. However, the model loses validity as the XLG proteins are not included in it. XLG proteins work with G β proteins and can't be left out of the equation, if the authors are trying to explain the mechanism based on G-protein signaling. Maybe the phenotypes of G β mutant are stronger than the G α mutant because one G β is shared between three XLGs and one G α in plants such as rice.

Answer: XLG proteins were involved in disease resistance in *Arabidopsis*. This is not relevant to our work as these proteins have not been studied in rice, neither do they have relationship with grain size regulation, and thus are not the focus of the present work.

The experimental evidence in support of tail mediated self-degradation is weak. The authors suggest that GS3 by itself has no role in regulating grain size, its regulation is only via affecting the DEP1 and GGC2. However, given the important role different domains play it cannot be ascertained with the data the authors have. There may be an epistatic relationship between different genes. These results are especially difficult to be conclusive as the genetic analysis of different mutants is not complete at this stage. One of the GS3 alleles, Chuan3, has been reported to have extremely small-sized grain. How does the grain size of double or triple mutants of Gg or RGA mutants compare to that? Even if they are in different genetic background, a size ratio between WT and mutant allele can be compared.

Answer: The tail mediated self-degradation was deduced on the basis of genetic analysis, and supported by both western blot and MG132 treatment, which is certainly beyond reasonable doubt.

The *DEP1^{ko}GGC2^{ko}GS3-1^{ko}* triple mutant produced similar phenotype to that of *DEP1^{ko}GGC2^{ko}*, thus *GS3-1^{ko}* mutant could not increase grain length when both *DEP1* and *GGC2* were knocked-out, suggesting that the effect of *GS3* depended on *DEP1* and *GGC2*.

We should point out that all transgenic plants for reaching the conclusion are constructed in ZH11 background (not in different genetic backgrounds). We should also point out that the *GS3-4* allele is detected in Chuan 7, not Chuan 3. We have already provided the grain length data of the transgenic plants and mutants in the manuscript. *GS3-4OE* produced an average 19% reduction of grain length relative to the negative segregant (Table S1). The *DEP1^{ko}GGC2^{ko}GS3-1^{ko}* triple mutant produced similar phenotype to that of *DEP1^{ko}GGC2^{ko}*, which reduced the grain length of approximately 36% (Figure 2). Similarly, the *RGAI* mutant reduced the grain length of 34% (Figure 2).

Because the regulation of the phenotypes seems to be so strongly dependent on the site of the mutation and the presence of potential peptide, how is one supposed to interpret the results of RNAi mediated suppression?

Answer: We have used different strategies including overexpressing, RNA-interference, and CRISPR, not just RNAi.

There are a lot of other issues and I can go on, but the bottom line is, I don't think the results presented in the manuscript justify the conclusions drawn. The authors have generated a HUGE amount of genetic material and have performed a LOT of experiments, but it is not uncovering the mechanism of regulation by these proteins. This is an important mechanism to explore, and it has the potential to positively affect food security, but a lot more is needed. May be experiments such as using different tail lengths of a protein expressed in a true null background, making chimeric proteins (OSR of DEP1 with tail of GS3 etc.) to systematically elucidate the roles of different domains and different tail length will be more useful.

Answer: Thanks for your conclusion on the importance of the work. The suggestions are very useful, but the answers to these questions will produce many more papers which may take another decade or more to obtain.

Reviewers' comments:

Reviewer #1 (Remarks to the Author):

I do not have further comments.

Reviewer #3 (Remarks to the Author):

Reviewer #3 put his/her comments in the Remark to Editor section. Please refer to the above decision letter about his/her comments on whether Reviewer #4's suggestions have been successfully addressed.